# ViT fine-tuning Vulnerability to label noise

## Abstract

Automatic annotation of large-scale datasets often introduces noisy labels, which degrade the performance of deep neural networks. Noisy Label Learning (NLL) has been well-studied for Convolutional Neural Networks (CNNs) however, its effectiveness for Vision Transformers (ViTs) remains less explored. In this work, namely `NLL-ViT`, we comprehensively benchmark the robustness of ViTs under diverse label noise settings, and also recommend an entropy based regularization to improve ViT performance. To this end, we address many key research questions including vulnerability of ViT to noisy labels, robustness of ViT relative to CNNs, effectiveness of existing NLL methods for ViT, correlation of prediction entropy reduction and ViT robustness, and the impact of recommended entropy regularization on robustness. In this study, we conducted more than 850 experiments, evaluating ViT-B/16 and ViT-L/16 fine-tuned via MLP-K on two standard loss functions and ten state-of-the-art NLL methods. Our benchmark spans three noise types including closed-set, open-set, and real-world, across eight datasets: CIFAR-10, CIFAR-100, CIFAR-10N, CIFAR-100N, CIFAR80-O, WebVision, Clothing1M, and Food-101N. Our findings show that ViT fine-tuning is vulnerable to noisy labels, ViTs fine-tuning is more robust to label noise compared to CNNs, existing CNN-based NLL methods are only effective in closed-set settings while failing to outperform standard losses under open-set and real-world noise settings. We also observe a strong correlation between prediction entropy reduction and ViT robustness. Also, the recommended entropy regularization combined with standard classification losses significantly enhances ViTs' robustness to noisy labels. We will release the `NLL-ViT` code publicly upon acceptance.

## 1 Introduction

Deep Neural Networks (DNNs) have achieved remarkable success across a wide range of computer vision tasks, largely due to the availability of large-scale annotated datasets Deng et al. (2009); Russakovsky et al. (2015); Lin et al. (2014); Shao et al. (2019). However, manual annotation of such datasets is both time-consuming and expensive. To scale dataset annotation, researchers have adopted alternatives such as automated labeling techniques Wang et al. (2016); Hu et al. (2016), web data crawling Xiao et al. (2015), and crowd-sourcing via platforms like Amazon Mechanical Turk Wei et al. (2022). These approaches, while cost-effective, often introduce *noisy labels*.

Developing learning strategies that are robust to label noise has become a key area of research. Existing work in noisy label learning (NLL) can be broadly categorized into four main directions: **(1)** *Label correction* methods that aim to detect and relabel noisy samples Zhang et al. (2017); Smart & Carneiro (2023); Li et al. (2022); Chen et al. (2024); Wang et al. (2024); **(2)** *Loss correction* methods that modify the loss function based on an estimated noise transition matrix Patrini et al. (2017); Reed et al. (2014); Han et al. (2018a); Wang et al. (2021); **(3)** *Refined training strategies* such as curriculum learning or co-teaching, designed to mitigate the influence of noisy data during training Tanaka et al. (2018); Ma et al. (2018); Jiang et al. (2018); Han et al. (2018b); Wu & Sun (2024); Yao et al. (2021); Karim et al. (2022); and **(4)** *Robust loss functions* that are inherently robust to the effects of noisy labels Ghosh et al. (2017); Ma et al. (2020); Wang et al. (2019); Zhang & Sabuncu (2018); Ye et al. (2023).

Given the success of Vision Transformers (ViTs) Dosovitskiy & et al. (2020) across a range of computer vision tasks Kirillov et al. (2023b); Radford et al. (2021); Yuan et al. (2021); Sharir et al. (2021), ViTs have emerged as the de facto standard in the field. Typically, ViTs are pre-trained on

✅ Accept Hypothesis  ❌ Reject Hypothesis

| Research Questions | Hypothesis | Closed-Set | | Open-Set | | Real-World | |
|---|---|---|---|---|---|---|---|
| | | p–value | Decision | p–value | Decision | p–value | Decision |
| **RQ₁:** Is ViT fine-tuning vulnerable to noisy labels? | $H_{10}$: ViT fine-tuning techniques are not vulnerable to training data label noise.
$H_{1A}$: ViT fine-tuning techniques are vulnerable to training data label noise. | 0.004 | ❌
✅ | 0.002 | ❌
✅ | 0.025 | ❌
✅ |
| **RQ₂:** Is ViT fine-tuning more robust to noisy labels compared to CNNs? | $H_{20}$: CNNs are more robust to label noise compared to ViT fine-tuning.
$H_{2A}$: ViT fine-tuning is more robust to label noise compared to CNNs. | 0.006 | ❌
✅ | 0.044 | ❌
✅ | 0.001 | ❌
✅ |
| **RQ₃:** Are existing NLL methods effective for ViT fine-tuning? | $H_{30}$: Existing NLL methods offer no advantage to ViT fine-tuning under noisy labels.
$H_{3A}$: Existing NLL methods significantly enhance ViT fine-tuning under noisy labels. | 0.006 | ❌
✅ | 0.469 | ✅
❌ | 0.174 | ✅
❌ |
| **RQ₄:** Is there a corelation between prediction entropy reduction and ViTs robustness to noisy labels? | $H_{40}$: There is no correlation between prediction entropy reduction and ViT robustness to noisy labels.
$H_{4A}$: There is a significant correlation between prediction entropy reduction and ViT robustness to noisy labels. | 0.004 | ❌
✅ | 0.002 | ❌
✅ | 0.003 | ❌
✅ |
| **RQ₅:** Can entropy regularization with standard classification losses improve the robustness of ViTs to noisy labels? | $H_{50}$: Explicit entropy regularization does not significantly improve the robustness of ViTs under noisy labels.
$H_{5A}$: Explicit entropy regularization improves the robustness of ViTs under noisy labels. | 0.002 | ❌
✅ | 0.007 | ❌
✅ | 0.011 | ❌
✅ |

Figure 1: **Overview of `NLL-ViT`.** The figure summarizes the hypotheses corresponding to each research question and indicates whether each hypothesis is accepted or rejected across three noise types: Closed-Set, Open-Set, and Real-World label noise for ViT-B/16 backbone.

large-scale datasets such as ImageNet-21K and then fine-tuned for downstream tasks rather than trained from scratch He et al. (2022); Kirillov et al. (2023a). Fine-tuning not only improves generalization with limited labeled data but also reduces computational costs. As a result, it has become standard practice in both vision and language domains Radford et al. (2021); Kirillov et al. (2023b); Yuan et al. (2021). While prior work has extensively explored the robustness of ViTs to adversarial attacks and out-of-distribution data Bai et al. (2021); Zhou et al. (2022); Paul & Chen (2022), their vulnerability to the presence of label noise training data remains relatively unexplored Liang et al. (2022). To address this very important issue, we present `NLL-ViT`. To the best of our knowledge, `NLL-ViT` is the first comprehensive benchmark designed to systematically evaluate the robustness of ViTs' fine-tuning under numerous label noise settings.

We benchmark two popular ViT backbones, ViT-B/16 and ViT-L/16, under three main categories of label noise including *closed-set*, *open-set*, and *real-world human noise*, and two subcategories, including instance-independent (Symmetric and Asymmetric) and instance-dependent (IDN-C and BadLabel). The evaluations span eight benchmark datasets including CIFAR-10, CIFAR-100, CIFAR80N-O, CIFAR-10N, CIFAR-100N, WebVision, Clothing1M, and Food-101N. We benchmark ten state-of-the-art NLL methods originally designed for CNNs, including GCE, SCE, NLNL, DivideMix, APL, NCE+AGCE, ANL, Robust DivideMix, CLIPCleaner, and NoiseGPT. We also benchmark two standard classification loss functions, including Cross-Entropy (CE) and Focal Loss (FL). `NLL-ViT` benchmark provides a unified framework to assess whether these techniques are effective when transferred to ViTs, offering new insights and highlighting key limitations. Specifically `NLL-ViT` performs more than 850 experiments to address five key research questions (RQ) as shown in Figure 1. Our core contributions are:

- We present `NLL-ViT`, a *first comprehensive benchmark* that systematically evaluates the robustness of ViTs under noisy labels, spanning three main categories: closed-set, open-set, and real-world label noise, and two subcategories: instance-independent noise including symmetric and Asymmetric, and instance-dependent noise including IDN-C and BadLabel. We conduct *more than 850 experiments* across eight benchmark datasets.

- We evaluate two ViT architectures and *ten state-of-the-art noisy label learning methods* to assess their effectiveness for ViTs. We also benchmark two standard classification losses including Cross-Entropy and Focal Loss. We observe that most existing NLL methods are only effective for ViT fine-tuning under closed-set label noise.

- We observe *a strong correlation between prediction entropy reduction and performance improvement*, highlighting a hidden functionality of NLL methods. Armed with this insight, we *recommend an entropy-based regularization* with standard classification loss functions to improve ViTs robustness to noisy labels.

## 2 RELATED WORK

Deep learning methods for Noisy Label Learning (NLL) are typically divided into four distinct categories: **Label Correction Methods:** These methods aim to identify and correct mislabeled data Xiao et al. (2015); Ko et al. (2023); Zhang et al. (2017); Zheng et al. (2021); Yi & Wu (2019); Zhang et al. (2024a); Wei et al. (2024). Li *et al.* Li et al. (2017b) average knowledge transfer from an expert model trained on a clean dataset to enhance a target model trained with noisy data. Recent works Wang et al. (2024); Feng et al. (2024) have utilized VLMs for correcting noisy labels. **Loss Correction Methods:** This category involves adjusting the loss function based on an estimated noise transition matrix Patrini et al. (2017); Reed et al. (2014); Han et al. (2018a); Sukhbaatar et al. (2014); Bae et al. (2024). Patrini *et al.* Patrini et al. (2017) developed loss correction techniques that are independent of the application domain and network architecture. Another approach, called 'Masking' Han et al. (2018a) uses human judgment to handle improbable label transitions effectively. **Refined Training Strategies:** These strategies are developed to adapt the training process for better handling of noisy labels Wang et al. (2018); Tanaka et al. (2018); Ma et al. (2018); Jiang et al. (2018); Han et al. (2018b); Kim et al. (2019); Ma et al. (2018); Zhang et al. (2024b); Li et al. (2020); Kim et al. (2024). Wang *et al.* Wang et al. (2018)specifically refine labels within a single training iteration by identifying and correcting mislabeled examples using a local outlier factor algorithm. Kim *et al.* Kim et al. (2019) have introduced a method known as Negative Learning for Noisy Labels (NLNL). Negative learning means an input sample does not belong to a class; instead of conventional Positive Learning (PL) where an input sample belongs to a class. NLNL does not provide wrong information to the model as frequently as PL and hence is more robust to noisy labels. **Robust Loss Functions:** These methods are specifically designed to mitigate the effects of noisy labels Ma et al. (2020); Ye et al. (2023); Wang et al. (2019); Zhang & Sabuncu (2018); Zhou et al. (2021); Amid et al. (2019a;b); Lyu & Tsang (2019). Generalized Cross Entropy (GCE) Zhang & Sabuncu (2018), for example, merges the benefits of Mean Absolute Error (MAE) and Cross-Entropy (CE). Symmetric Cross Entropy (SCE) Wang et al. (2019) addresses noisy data by combining Reverse Cross Entropy (RCE) with CE, where RCE is defined as: $-\sum_{k=1}^{k_c} \mathbf{p}(k|\mathbf{x}_i) \log \mathbf{q}(k|\mathbf{x}_i)$. Zhou *et al.* Zhou et al. (2021) proposed Asymmetric Generalized Cross Entropy (AGCE) fulfilling the noise tolerance condition proposed by Ghosh *et al.* Ghosh et al. (2017). Ma *et al.* Ma et al. (2020) designed Active Passive Loss (APL), which integrates an active component that assigns high probability to the ground truth class and a passive component that diminishes the likelihood of high probabilities for other classes. One implementation of APL is NCE+RCE, which has proven effective in noisy conditions. Expanding on this concept, Ye *et al.* Ye et al. (2023), noting that existing passive loss functions are scaled versions of MAE, proposed a new class of passive loss functions called Normalized Negative Loss Functions (NNLFs). An example of NNLF is ANL-CE loss which combines NCE with negative normalized cross entropy (NNCE).

## 3 BENCHMARK DESIGN

**Noisy Label Learning.** In supervised learning, the objective is to learn a mapping from input features $\mathbf{x} \in \mathcal{X}$ to labels $y \in \mathcal{Y}$ using a training dataset $\mathcal{D} = \{(\mathbf{x}_i, y_i)\}_{i=1}^N$. Typically, labels $y_i$ are assumed to be accurate, reflecting the true class of each instance $\mathbf{x}_i$. However, real-world datasets often contain label noise, where observed labels $\tilde{y}_i$ may differ from true labels $y_i$. Noisy label learning seeks to train robust models despite this noise, minimizing the impact of incorrect labels on performance. Formally, let $\mathcal{D}_{\text{clean}} = \{(\mathbf{x}_i, y_i)\}_{i=1}^n$ represent a clean dataset with true labels, and $\mathcal{D}_{\text{noisy}} = \{(\mathbf{x}_i, \tilde{y}_i)\}_{i=1}^n$ denote the observed noisy dataset. The probability of observing a noisy label is modeled as: $P(\tilde{y}_i \mid \mathbf{x}_i, y_i) = \mathbf{T}(\mathbf{x}_i, y_i)$, where $\mathbf{T}$ is the noise transition matrix, which may depend on the instance $\mathbf{x}_i$, the true label $y_i$, or both. The goal is to learn a classifier $f : \mathcal{X} \to \mathcal{Y}$ that generalizes well to the true label distribution ($\mathcal{Y}$) despite training on $\mathcal{D}_{\text{noisy}}$.

**Label Noise Categories.** Label noise is mainly categorized into closed-set and open-set synthetic label noise, and real-world human label noise. In *closed-set noise*, labels $\tilde{y}_i$ are restricted to the known

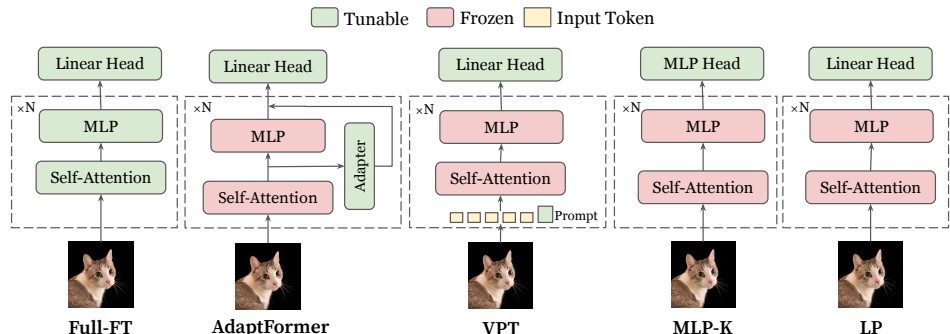

Figure 2: Comparison of Five Fine-Tuning Strategies for Vision Transformers. Details of architectural modifications for Full Fine-Tuning, AdaptFormer Chen et al. (2022), Visual Prompt Tuning Jia et al. (2022), MLP-K, and Linear Probing He et al. (2020).

set $\mathcal{Y}$, i.e., the true label $y_i$ is replaced with another class in $\mathcal{Y}$. It is divided into instance-independent and instance-dependent noise. We benchmark two instance-independent types: symmetric or uniform noise, where all incorrect labels in $\mathcal{Y} \setminus y_i$ are equally likely, and asymmetric or class-conditional noise, where some confusions are favored (e.g., cat $\leftrightarrow$ dog). For instance-dependent noise, we use IDN-C Ye et al. (2023), which trains a model to identify confusing samples, and BadLabel Zhang et al. (2024b), an adversarial method that produces the most challenging closed-set noise. See Supplementary Section A for details. *Open-set noise* commonly arises in data from diverse sources (e.g., web scraping), where samples from unknown classes are mislabeled as known ones Wan et al. (2024). Formally, let $\mathcal{X}_{\text{open}}$ denote the space of unknown classes, with $x'_j \in \mathcal{X}_{\text{open}}$ not belonging to any class in $\mathcal{Y}$. The open-set noisy dataset is then $\mathcal{D}_{\text{OSN}} = \mathcal{D}_{\text{clean}} \cup (x'_j, \tilde{y}_j)_{j=1}^m$, where $\tilde{y}_j \in \mathcal{Y}$. *Real-world human label noise* Xiao et al. (2015); Li et al. (2017a); Lee et al. (2018) arises from annotator errors and shows complex, non-uniform patterns influenced by expertise, fatigue, ambiguous guidelines, or subjectivity. It is common in datasets built from crowd-sourcing, expert labeling, or user-generated data.

**Noise Settings.** For closed-set noise, we follow standard protocols on CIFAR-10 and CIFAR-100 Ma et al. (2020); Ye et al. (2023); Zhou et al. (2021). Symmetric noise flips labels randomly with $\eta \in \{0.4, 0.6, 0.8\}$, while asymmetric noise flips with similar confusing classes with $\eta \in \{0.2, 0.3, 0.4\}$ (e.g., TRUCK→AUTOMOBILE, CAT↔DOG for CIFAR-10, and circular flips within 20 superclasses for CIFAR-100) Wang et al. (2019); Zhang & Sabuncu (2018). We also include IDN-C Chen et al. (2021) and BadLabel Zhang et al. (2024b) at $\eta \in \{0.6, 0.8\}$. For open-set noise, the last 20 CIFAR-100 classes are treated as out-of-distribution and injected into the remaining 80 classes, with additional in-distribution corruption using symmetric/asymmetric noise, giving a total noise ratio $\eta_o = 0.2 + 0.8\eta_c$. For real-world human noise, we evaluated on CIFAR-10N, CIFAR-100N, WebVision Li et al. (2017a), Clothing1M Xiao et al. (2015), and Food-101N Lee et al. (2018). See Supplementary Section B for additional details.

**Vision Transformer Fine-tuning Paradigm.** Fine-tuning adapts pre-trained models on large-scale datasets like ImageNet Deng et al. (2009) to downstream tasks by optimizing parameters on task-specific data $\mathcal{D}_{\text{task}} = (\mathbf{x}_i, y_i)_{i=1}^n$ with loss $\mathcal{L}$. **Full Fine-Tuning** (Full-FT) updates all parameters but is computationally heavy. More efficient alternatives include **Linear Probing** (LP), which tunes only the final layer, and **MLP-K**, which fine-tunes a lightweight classification head. **AdaptFormer** Chen et al. (2022) inserts adapter modules for task-specific adaptation, while **Visual Prompt Tuning** (VPT) Jia et al. (2022) prepends learnable prompts $P = \{p_1, \ldots, p_m\}$ to the input sequence. A visual comparison of these techniques is provided in Figure 2.

**Evaluated Baselines** We evaluate on cross-entropy (CE) and Focal-Loss (FL) as standard classification loss functions (SLF). Apart form SLF, we also evaluate on ten SOTA NLL methods, originally developed for CNNs, to benchmark their performance on ViTs including: Generalized Cross-Entropy (GCE) Zhang & Sabuncu (2018), Symmetric Cross-Entropy (SCE) Wang et al. (2019), Negative Learning for Noisy Labels (NLNL) Kim et al. (2019), DivideMix Li et al. (2020), Active-Passive Losses (APL) combining Normalized Cross-Entropy and Reverse

Table 1: **RQ1:** Average performance across clean and noisy settings for closed-set, open-set, and real-world noise using five fine-tuning techniques. Paired t-test (t-values and p-values) reports the significance of degradation. AdF: AdaptFormer.

| Noise | Dataset | Full-FT | AdF | VPT | MLP-K | LP | Average | p-value |
|---|---|---|---|---|---|---|---|---|
| **Closed-Set** | Clean | **95.44** | 90.55 | 93.61 | 91.46 | 91.34 | **92.48** | 0.004 |
| | Noisy | 40.06 | 65.25 | 62.00 | 65.54 | **66.05** | 59.78 | |
| **Open-Set** | Clean | 90.68 | **92.12** | 91.28 | 89.72 | 89.45 | **90.65** | 0.002 |
| | Noisy | 40.75 | 55.01 | 54.47 | **64.97** | 62.79 | 55.60 | |
| **Real-World** | Clean | **95.44** | 90.55 | 93.61 | 91.46 | 91.34 | **92.48** | 0.025 |
| | Noisy | 58.33 | 78.62 | 79.46 | **81.21** | 77.81 | 75.09 | |

Table 2: Comparisons on close-set label noise averaged on CIFAR-10 and CIFAR-100. Standard loss functions (SLF) performance is averaged over cross entropy and Focal loss. Noisy label learning methods (NLL) performance is averaged over GCE, SCE, NLNL, DivideMix, NCE+RCE, NCE+AGCE, ANL-CE, Robust DivideMix, CLIPCleaner, and NoiseGPT. H is the recommended entropy regularization.

| Architecture | Symmetric | | | Asymmetric | | | IDN-C | | BadLabel | | Average | p-value |
|---|---|---|---|---|---|---|---|---|---|---|---|---|
| | 0.4 | 0.6 | 0.8 | 0.2 | 0.3 | 0.4 | 0.6 | 0.8 | 0.6 | 0.8 | | |
| RQ2: Performance comparison of CNN and ViT with CE. | | | | | | | | | | | | |
| CNN+CE | 71.34 | 65.67 | 38.84 | 75.88 | 69.18 | 57.38 | 53.94 | 28.19 | 27.28 | 7.15 | 49.48 | 0.006 |
| ViT-S/16+CE | 77.31 | 70.67 | 49.12 | 77.69 | 73.19 | 67.56 | 51.03 | 31.81 | 31.18 | 9.63 | 53.92 | |
| RQ3: Performance comparison of ViTs+SLF and ViTs+NLL. | | | | | | | | | | | | |
| ViT-B/16+SLF | 76.09 | 55.39 | 31.25 | 82.69 | 76.55 | 68.27 | 54.19 | 29.30 | 37.50 | 12.76 | 52.39 | 0.006 |
| ViT-B/16+NLL | 84.98 | 81.69 | 70.88 | 86.23 | 82.73 | 75.84 | 65.90 | 42.12 | 44.04 | 19.08 | 65.35 | |
| RQ5: Performance comparison of ViT+SLF and ViT+SLF+H, the entropy regularization. | | | | | | | | | | | | |
| ViT-B/16+SLF | 76.09 | 55.39 | 31.25 | 82.69 | 76.55 | 68.27 | 54.20 | 29.30 | 37.50 | 12.76 | 52.39 | 0.002 |
| ViT-B/16+SLF+$H$ | 89.36 | 85.86 | 71.51 | 89.23 | 87.50 | 83.50 | 70.07 | 43.60 | 42.48 | 16.79 | 67.99 | |

Cross-Entropy (NCE+RCE) Ma et al. (2020), NCE with Asymmetric Generalized Cross-Entropy (NCE+AGCE) Zhou et al. (2021), Active Noisy Label learning with Cross-Entropy (ANL-CE) Ye et al. (2023), RobustDivideMix Zhang et al. (2024b), CLIPCleaner Feng et al. (2024), and NoiseGPT Wang et al. (2024). The choice of SOTA NLL methods is constrained by the availability of open-source code and their compatibility with ViT architectures, ensuring seamless integration. For each method, the hyperparameters recommended in the respective papers were employed.

## 4 RESEARCH QUESTIONS

In `NLL-ViT`, we address five key research questions regarding the vulnerability of ViTs' fine-tuning to noisy labels. For each research question (RQ), we formulate the corresponding null hypotheses ($H_{iO}$) and alternative hypotheses ($H_{iA}$). We evaluated the statistical significance of our findings using the paired t-test. Figure 1 outlines these research questions along with their respective hypotheses.

Table 3: Average performance comparison on CIFAR80N-O under open-set label noise. SLF performance is averaged over CE and FL. NLL performance is averaged over ten SOTA methods.

| Architecture | Symmetric | | Asymmetric | Average | p-value |
|---|---|---|---|---|---|
| | 0.2 | 0.8 | 0.4 | | |
| RQ2: Performance comparison of CNN and ViT with CE. | | | | | |
| CNN+CE | 42.26 | 12.1 | 22.68 | 25.68 | 0.044 |
| ViT-S/16+CE | 73.18 | 25.89 | 46.52 | **48.54** | |
| RQ3: Performance comparison of ViTs+SLF and ViTs+NLL. | | | | | |
| ViT-B/16+SLF | 83.59 | 38.48 | 69.72 | 63.93 | 0.469 |
| ViT-B/16+NLL | 79.53 | 72.07 | 71.32 | **74.31** | |
| RQ5: Performance comparison of ViT+SLF and ViT+SLF+H. | | | | | |
| ViT-B/16+SLF | 83.59 | 38.48 | 69.72 | 63.93 | 0.007 |
| ViT-B/16+SLF+$H$ | 88.09 | 76.17 | 72.65 | **78.97** | |
| ViT-L/16+SLF | 80.08 | 26.76 | 57.03 | 54.62 | |
| ViT-L/16+SLF+$H$ | 87.89 | 78.51 | 66.21 | 77.47 | |

Table 4: Average performance comparison on CIFAR10N, CIFAR100N, WebVision, Clothing1M, and Food101N real-world noisy label datasets. SLF performance is averaged over CE and FL. NLL performance is averaged over ten SOTA methods.

| Architecture | CIFAR10N | | | | | CIFAR100N | WebVision | Clothing1M | Food101N | Average | p-value |
|---|---|---|---|---|---|---|---|---|---|---|---|
| | Aggre | Rand1 | Rand2 | Rand3 | Worst | | | | | | |
| RQ2: Performance comparison of CNN and ViT with CE. | | | | | | | | | | | |
| CNN+CE | 87.77 | 85.02 | 86.46 | 85.16 | 77.69 | 55.50 | 61.20 | 69.21 | 84.51 | 71.65 | 0.001 |
| ViT-S/16+CE | 94.53 | 95.31 | 94.53 | 93.75 | 90.23 | 68.75 | 88.47 | 64.94 | 78.31 | **73.77** | |
| RQ3: Performance comparison of ViTs+SLF and ViTs+NLL. | | | | | | | | | | | |
| ViT-B/16+SLF | 94.53 | 95.31 | 94.34 | 93.95 | 90.23 | 68.17 | 88.57 | 63.58 | 76.51 | **85.01** | 0.174 |
| ViT-B/16+NLL | 94.62 | 94.71 | 94.21 | 94.52 | 92.27 | 64.41 | 74.52 | 62.62 | 81.87 | 81.87 | |
| RQ5: Performance comparison of ViT+SLF and ViT+SLF+H, the entropy regularization. | | | | | | | | | | | |
| ViT-B/16+SLF | 94.53 | 95.31 | 94.34 | 93.95 | 90.23 | 68.17 | 88.57 | 63.58 | 76.51 | 85.01 | 0.011 |
| ViT-B/16+SLF+$H$ | 95.31 | 96.09 | 95.12 | 95.12 | 93.17 | 73.24 | 89.26 | 67.01 | 76.97 | **86.81** | |

RQ1. Is ViT fine-tuning vulnerable to noisy labels?

**Motivation.** While the robustness of ViTs has been extensively studied in adversarial and out-of-distribution (OOD) settings Bai et al. (2021); Zhou et al. (2022); Paul & Chen (2022), their robustness to noisy labels remains less explored. This research question seeks to fill in this gap by investigating how various ViT fine-tuning techniques perform under different categories of label noises.

**Experimental design.** Experiments are performed on the three label noise categories and five fine-tuning techniques for ViTs including Full-FT, AdaptFormer, VPT, MLP-K, and LP as discussed in Section 3. In this question, clean training data labels are required to measure the degradation in ViT fine-tuning performance due to noisy labels. Therefore, datasets such as Clothing1M, Food-101N, and WebVision that lack clean training sets are not applicable.

**Comparison across fine-tuning techniques.** Table 1, shows a comparison of five ViT finetuning techniques on clean and noisy datasets. For *closed-set noise*, both clean and noisy performances are averaged over CIFAR-10 and CIFAR100. The noisy performance is averaged over six noise settings in both datasets $\eta_{\text{sym}} \in \{0.4, 0.6, 0.8\}$ and $\eta_{\text{asym}} \in \{0.2, 0.3, 0.4\}$. In this noise category, Full-FT has performed best on clean datasets while LP achieved best performance under noisy label settings. In *open-set noise* category, clean performance is reported on CIFAR80N-O, while noisy performance is averaged over three noise settings $\eta_o = 0.2 + 0.8\eta_{sym}$ where $\eta_{\text{sym}} \in \{0.2, 0.8\}$, and $\eta_o = 0.2 + 0.8\eta_{\text{asym}}$ for $\eta_{\text{asym}} = 0.4$. In this noise category, Adaptformer performed best in clean while MLP-K achieved best performance under noisy settings. In *real-world label noise* clean performance is averaged over CIFAR10 and CIFAR100, while noisy performance is averaged over CIFAR10N and CIFAR100N. Full-FT obtained the best performance in clean dataset while MLP-K achieved the best performance under noisy settings. For detailed results, see Supp. Section C.1.

**ViT performance degradation.** All fine-tuning techniques experience significant performance degradation under noisy settings. Full-FT suffers the largest accuracy decline which may be attributed to the distortion caused by noisy labels in the learned feature space. Our findings are consistent with previous studies Kumar et al. (2022),Hua et al. (2023). MLP-K and LP have emerged as the most robust fine-tuning techniques in noisy label settings, probably because of fewer tunable parameters. We observe a similar trend for ViT-L/16 (Supp. Section D.1).

**Statistical significance.** To statistically assess the vulnerability of ViT fine-tuning to label noise, we performed the paired t-test comparing the performance of each fine-tuning technique in clean versus noisy settings. The test yields a p-value of 0.004, 0.002, 0.025 for the three noise categories, which are below the significance threshold of 0.05. Therefore, the null hypothesis ($H_{1O}$) is rejected and alternate hypothesis ($H_{1A}$: *ViT fine-tuning techniques are vulnerable to training data label noise*) is accepted.

RQ2: Is ViT fine-tuning more robust to noisy labels compared to CNNs?

**Motivation.** CNNs have been extensively explored in the context of NLL, with many existing methods specifically designed around CNN architectures. In contrast, ViTs leverage self-attention mechanisms, which may offers distinct robustness properties Bai et al. (2021); Zhou et al. (2022); Paul & Chen (2022). This research question aims to directly compare ViTs and CNNs under vary-

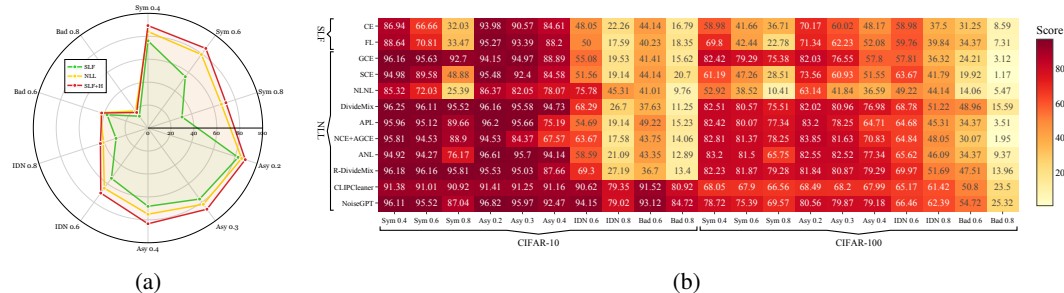

(a)                                                                          (b)

Figure 3: **RQ3:** a) Average performance of ViT+SLF and ViT+NLL under closed-set label noise on CIFAR-10 and CIFAR-100 with the ViT-B/16 backbone. Noise types include: Sym (Symmetric) Ye et al. (2023), Asym (Asymmetric) Ye et al. (2023), IDN (IDN-C) Chen et al. (2021), and Bad (BadLabel) Zhang et al. (2024b). b) Detailed results across datasets and noise settings.

ing types of noisy-label conditions to assess their relative robustness and guide model selection in practical settings.

**Experimental Design.** We evaluate the robustness of CNNs and ViTs under the three noise categories introduced in Section 3. Both architectures are trained with cross-entropy loss and fine-tuned using MLP-K fine-tuning to ensure a fair comparison. For CNNs, we use an ImageNet-pretrained ResNet-50 He et al. (2016) backbone, while for ViTs we adopt an ImageNet-pretrained ViT-S/16 Bai et al. (2021). This choice of backbones ensures comparability, as ResNet-50 and ViT-S/16 are similar in size, with 25M and 22M parameters, respectively.

**ViTs and CNNs Performance Comparison.** Table 2 reports the average performance of CNNs and ViTs under closed-set noise on CIFAR-10 and CIFAR-100. Across both datasets, ViTs consistently outperform CNNs. Table 3 presents results under open-set label noise across three different noise settings. ViTs again demonstrate a clear advantage, achieving significantly higher accuracy than CNNs in all cases. Table 4 compares both architectures on five real-world noisy label datasets. On average, ViTs achieve superior performance, further confirming their robustness to label noise. See Supplementary E for the detailed result on ViT-S/16. Similar performance trends are observed for larger ViT variants, including ViT-B/16 (Supplementary Section C.2) and ViT-L/16 (Supplementary Section D.2).

**Statistical significance.** The statistically significance of RQ2 on three noise categories is evaluated using paired t-test. The test yields p-values of 0.006, 0.044, and 0.001 for the three noise categories, which are below the significance threshold of 0.05. Therefore, the null hypothesis ($H_{2O}$) is rejected and alternate hypothesis ($H_{2A}$: *ViT fine-tuning is more robust to label noise compared to CNNs*) is accepted.

RQ3: ARE EXISTING NLL METHODS EFFECTIVE FOR ViT FINE-TUNING?

**Motivation.** Most existing SOTA NLL methods have been proposed and validated only on CNN. However, ViTs differ fundamentally from CNNs in inductive biases and representation learning strategies. This raises an important question: can NLL methods designed for CNNs be effectively applied to ViTs, or is there a need for ViT-specific solution?

**Experimental Design.** To investigate this RQ, we apply ten SOTA CNN-based NLL methods to ViT-MLP-K fine-tuning across three noise categories (Section 3). Each method's effectiveness is benchmarked against ViT baselines trained with standard classification losses (SLF) including CE and FL, to assess whether these NLL methods are effective for ViTs.

**Comparison of ViT with and without NLL.** Figure 3 and 4 benchmark the ViT-MLP-K for SLF and SOTA NLL methods across three noise categories. Table 2 compares the average performance of ViTs with and without NLL methods under closed-set noise using the CIFAR-10 and CIFAR-100 datasets. Across all 10 noise settings, ViT+NLL consistently outperforms ViT+SLF by a significant margin, demonstrating the effectiveness of NLL methods for this noise category. Table 3, compares the same performance for open-set noise under three different noise settings. For smaller symmetric

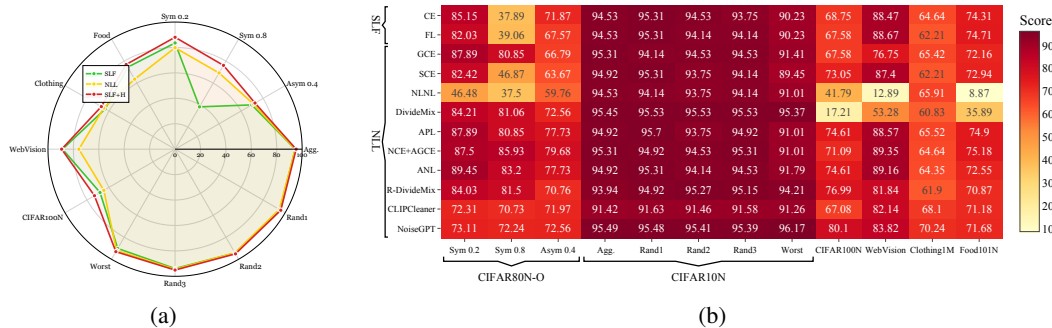

|  | Sym 0.2 | Sym 0.8 | Asym 0.4 | Agg. | Rand1 | Rand2 | Rand3 | Worst | CIFAR100N | WebVision | Clothing1M | Food101N |
|---|---|---|---|---|---|---|---|---|---|---|---|---|
| CE | 85.15 | 37.89 | 71.87 | 94.53 | 95.31 | 94.53 | 93.75 | 90.23 | 68.75 | 88.47 | 64.64 | 74.31 |
| FL | 82.03 | 39.06 | 67.57 | 94.53 | 95.31 | 94.14 | 94.14 | 90.23 | 67.58 | 88.67 | 62.21 | 74.71 |
| GCE | 87.89 | 80.85 | 66.79 | 95.31 | 94.14 | 94.53 | 94.53 | 91.41 | 67.58 | 76.75 | 65.42 | 72.16 |
| SCE | 82.42 | 46.87 | 63.67 | 94.92 | 95.31 | 93.75 | 94.14 | 89.45 | 73.05 | 87.4 | 62.21 | 72.94 |
| NLNL | 46.48 | 37.5 | 59.76 | 94.53 | 94.14 | 93.75 | 94.14 | 91.01 | 41.79 | 12.89 | 65.91 | 8.87 |
| DivideMix | 84.21 | 81.06 | 72.56 | 95.45 | 95.53 | 95.53 | 95.53 | 95.37 | 17.21 | 53.28 | 60.83 | 35.89 |
| APL | 87.89 | 80.85 | 77.73 | 94.92 | 95.7 | 93.75 | 94.92 | 91.01 | 74.61 | 88.57 | 65.52 | 74.9 |
| NCE+AGCE | 87.5 | 85.93 | 79.68 | 95.31 | 94.92 | 94.53 | 95.31 | 91.01 | 71.09 | 89.35 | 64.64 | 75.18 |
| ANL | 89.45 | 83.2 | 77.73 | 94.92 | 95.31 | 94.14 | 94.53 | 91.79 | 74.61 | 89.16 | 64.35 | 72.55 |
| R-DivideMix | 84.03 | 81.5 | 70.76 | 93.94 | 94.92 | 95.27 | 95.15 | 94.21 | 76.99 | 81.84 | 61.9 | 70.87 |
| CLIPCleaner | 72.31 | 70.73 | 71.97 | 91.42 | 91.63 | 91.46 | 91.58 | 91.26 | 67.08 | 82.14 | 68.1 | 71.18 |
| NoiseGPT | 73.11 | 72.24 | 72.56 | 95.49 | 95.48 | 95.41 | 95.39 | 96.17 | 80.1 | 83.82 | 70.24 | 71.68 |

(a)                  (b)

Figure 4: **RQ3:** a) Average performance of ViT+SLF and ViT+NLL with ViT-B/16 under open-set noise (CIFAR80N-O) and real-world noisy datasets: CIFAR-10N, CIFAR-100N, WebVision, Food101N, and Clothing1M. b) Detailed results across datasets and noise settings.

noise ViT+SLF have better performance than ViT+NLL. However for larger noise rates ViT+NLL has performed better. Table 4 shows the comparison on five real-world noisy label datasets. On average, ViT+SLF performs better than ViT+NLL, suggesting that existing NLL methods are not effective for ViTs under real-world human label noise settings. A similar trend is observed for ViT-L/16 (See Supplementary Section D.3).

**Statistical significance.** For the closed-set noise category, the p-value (0.006) is below the significance threshold of 0.05, leading us to reject the null hypothesis ($H_{3O}$) and accept the alternative hypothesis ($H_{3A}$: *Existing NLL methods significantly enhance ViT fine-tuning under label noise*). However, for open-set and real-world noise, the p-values exceed the threshold, so we accept the null hypothesis ($H_{3O}$: *Existing NLL methods offer no advantage for ViT fine-tuning under label noise*). These results indicate that while existing CNN-based NLL methods can enhance robustness in ViTs under closed-set noise, they are not effective in more complex noise scenarios like open-set and real-world human label noise.

RQ4: IS THERE A CORRELATION BETWEEN PREDICTION ENTROPY REDUCTION WITH EPOCHS AND VITS ROBUSTNESS TO NOISY LABELS?

**Motivation.** Prediction entropy over the entire dataset $\mathcal{X}$ is defined as: $H(\mathcal{X}) = -\frac{1}{n}\sum_{i=1}^{n}\sum_{k=1}^{c} p(k \mid \mathbf{x}_i) \log p(k \mid \mathbf{x}_i)$, where $p(k \mid \mathbf{x}_i)$ is the softmax prediction probability. It reflects a model's confidence, where higher values typically indicate uncertainty or confusion, often caused by noisy labels. Although NLL methods have been well studied, this relationship has not been investigated in literature. In this RQ, we systematically study the relationship of entropy reduction and model robustness for ViTs. We define prediction entropy reduction as: $\Delta H = (H_1 - H_{e_t})/H_1$, where $H_1$ is the entropy computed after the first epoch, and $H_{e_t}$ is the entropy after the $e_t$ epochs.

**Experimental design.** We investigate this relationship using SLF and ten SOTA NLL methods across three noise categories using eight datasets. We then conduct a Pearson correlation analysis between prediction entropy reduction and model accuracy across all methods and noise settings.

**Prediction entropy relates to model robustness.** Our Pearson correlation analysis demonstrates a strong positive relationship between prediction entropy reduction and the robustness of ViTs across all three noise categories with Pearson correlation coefficients of 0.76, 0.79, and 0.77 (Figure 5). This suggests that models exhibiting greater reduction in prediction entropy from early to late training stages tend to achieve higher test accuracy under label noise.

**Statistical significance.** We computed the Pearson correlation between test accuracy and prediction entropy reduction across the three noise categories and assessed the statistical significance of these correlations using the Student's t-test for correlation coefficients. The analysis yielded p-values of 0.0004, 0.002, and 0.003, respectively. Since all p-values are below the significance threshold of 0.05, we reject the null hypothesis ($H_{4O}$) and accept the alternative hypothesis ($H_{4A}$: *There is a significant correlation between prediction entropy reduction and ViT robustness to noisy labels*).

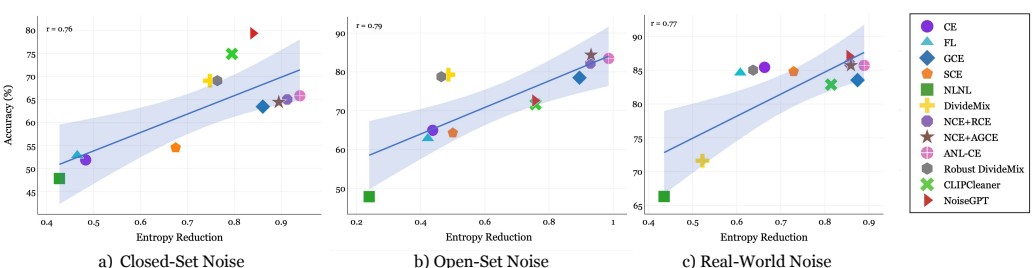

a) Closed-Set Noise      b) Open-Set Noise      c) Real-World Noise

Figure 5: **RQ4:** Entropy Reduction vs. Accuracy across two SLFs and ten SOTA NLL methods (ViT-B/16 backbone).

### RQ5: CAN ENTROPY REGULARIZATION WITH SLF IMPROVE THE ROBUSTNESS OF ViTS TO NOISY LABELS?

**Motivation.** The findings from RQ4 serve as the primary motivation for RQ5, where we observed a strong positive correlation between prediction entropy reduction and ViT robustness. This suggests that prediction entropy reduction during training may improve robustness of ViTs to label noise. Building on this insight, we investigate whether explicitly incorporating entropy regularization with SLFs can enhance the robustness of ViTs to noisy label supervision.

**Experimental Design.** We augment SLFs with an entropy regularization term, forming the composite loss: $\mathcal{L} = \mathcal{L}_{\text{SLF}} + \lambda H$, where $\mathcal{L}_{\text{SLF}} \in \{\text{CE}, \text{FL}\}$ and $H$ is the average prediction entropy. The regularization coefficient $\lambda$ is linearly increased from 0 to 0.3 over the course of training to progressively enforce confident predictions. An ablation study on different choices of $\lambda$ values is included in the Supplementary Section C.5.

**Entropy regularization improves ViTs robustness.** The intuitive proof of entropy regularization improves robustness is provided in Supp. Section G. Tables 2, 3, and 4 compares the average performance of ViTs with and without entropy regularization under three different noise categories. In all experiments, ViT+SLF+$H$ consistently outperforms ViT+SLF by a significant margin, demonstrating the effectiveness of entropy regularization. A similar trend is observed for ViT-S/16 (Supp. Section E), ViT-L/16 (Supp. Section D.5), and ResNet-50 (Supp. Section F).

**Statistical Significance.** Paired t-test is performed to assess the statistical significance of the performance improvement by entropy regularization, resulting in p-values of 0.0002, 0.007, and 0.011 for three noise categories. For all noise, categories p-values fall below the significance threshold of 0.05, leading us to reject the null hypothesis ($H_{5O}$) and accept the alternative hypothesis ($H_{5A}$: *Explicit entropy regularization improves the robustness of ViTs under noisy labels*). The p-value for open-set is computed using both ViT-B/16 and ViT-L/16 using CE and FL losses to increase the number of samples.

## 5 CONCLUSION

We evaluate the vulnerability of Vision Transformer (ViT) fine-tuning to noisy labels in training data, focusing on three major noise categories: closed-set, open-set, and real-world human annotation noise. Our study spans eight benchmark datasets and includes two SLFs along with ten SOTA noisy label learning methods. Our presented `NLL-ViT` benchmark is a large-scale systematic evaluation of the vulnerability of ViTs under diverse label noise settings. `NLL-ViT` is a large-sclae benchmark that systematically evaluate the ViTs vulnerability to label noise through five core research questions. Our findings reveal that ViT fine-tuning is vulnerable to noisy labels (RQ1), ViTs are more robust to noisy labels than CNNs (RQ2), existing NLL methods offer limited effectiveness for ViTs beyond closed-set noise (RQ3), prediction entropy reduction strongly correlates with ViTs robustness to label noise (RQ4), and recommended entropy regularization with SLFs significantly enhances ViT performance under noisy labels (RQ5). These insights highlight the need for ViT-specific noisy label learning strategies to be developed. In addition, the recommended entropy-based regularization is a simple yet effective way to enhance ViTs robustness to label noise.

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

VIT FINE-TUNING VULNERABILITY TO LABEL NOISE

SUPPLEMENTARY MATERIAL



## LLM USAGE STATEMENT

We made limited use of Large Language Models (LLM) to enhance the clarity and readability of the text. LLMs were not involved in the conception of ideas, experiment design, analysis, or the production of results.

## REPRODUCIBILITY STATEMENT

We have taken several steps to ensure the reproducibility of our work:

- **Code and Implementation:** We will release our full codebase, including training and evaluation scripts, upon publication. The implementation uses standard deep learning frameworks (PyTorch) with widely available dependencies.

- **Datasets:** All datasets used in this work are publicly available: CIFAR-10/100, CIFAR-10N/100N, CIFAR80N-O, WebVision, Food101N, and Clothing1M. We provide preprocessing instructions where applicable.

- **Hyperparameters:** We report or cite the sources which we followed for key hyperparameters (learning rate, optimizer, batch size, weight decay, etc.) We have provided optimal weight for entropy regularization weight in the main text.

- **Experimental Setup:** Experiments were conducted on standard 4x NVIDIA A14 GPU machine. We provide details on training epochs, noise settings, and random seeds to facilitate exact replication.

- **Evaluation:** Results are averaged over three independent runs with different random seeds, and we report mean and standard deviation.

We believe these steps make it straightforward for other researchers to reproduce and extend our results.

The supplementary document is organized as follows: Section A provides additional details on closed-set synthetic label noise. Section B provides additional details on datasets and implementation. Section C presents a detailed breakdown of the averaged results reported in the main paper, organized by each research question. Section D reports the results for the ViT-L/16 backbone corresponding to all five research questions. Section E presents results on ViT-S/16 backbone and Section F presents results on ResNet-50 backbone. Section G provide intuitive proof with worked example as to how entropy regualrization improves robustness.

## A  CLOSED-SET SYNTHETIC LABEL NOISE

In closed-set noise, noisy labels $\tilde{y}_i$ are restricted to the known label set $\mathcal{Y}$ i.e. noise corrupts the true label $y_i$ by assigning another class within $\mathcal{Y}$: $\tilde{y}_i \in \mathcal{Y}$. Closed-set noise is further divided into instance-independent and instance-dependent noise, based on whether the noise depends on instance features $\mathbf{x}_i$ or not.

### A.1  INSTANCE-INDEPENDENT LABEL NOISE

Instance-independent label noise depends only on the true label $y_i$. The noise transition probability is modeled as: $P(\tilde{y}_i \mid y_i) = \mathbf{T}(y_i)$. Two instance-independent noises are commonly used:

**Symmetric or Uniform Noise** Ye et al. (2023) assumes a constant noise transition probability across $\mathcal{Y}$. All labels are equally likely to be transitioned to any incorrect label in $\mathcal{Y} \setminus \{y_i\}$. For the $j$-th class, noise rate ($\eta_j$) is defined as the ratio of the number of incorrect labels in that class ($l_j$) to the number of total samples of that class ($n_j$): $\eta_j = l_j/n_j$. In symmetric noise $n_j$ is same for all classes

and is represented as $\eta$. For $|\mathcal{Y}| = c$ classes , the transition probability is:

$$P(\tilde{y}_i = k \mid y_i = j) = \begin{cases} 1 - \eta, & \text{if } k = j, \\ \frac{\eta}{c-1}, & \text{if } k \neq j. \end{cases} \tag{1}$$

**Asymmetric or Class-Conditional Noise** Ye et al. (2023) allows certain incorrect labels to be favored, reflecting real-world biases or annotation errors for example a cat is more likely to be mislabeled as dog instead of truck. The error rate $\eta_j$ will vary across different class labels. The transition probabilities are: $P(\tilde{y}_i = k \mid y_i = j) = T_{jk}$, where $T_{jk} \neq T_{kj}$ for $j \neq k$, and $\sum_{k \in \mathcal{Y}} T_{jk} = 1$.

## A.2 INSTANCE-DEPENDENT LABEL NOISE

Instance-dependent label noise varies with both the true label $y_i$ and the instance features $\mathbf{x}_i$: $P(\tilde{y}_i \mid \mathbf{x}_i, y_i) = \mathbf{T}(\mathbf{x}_i, y_i)$, meaning certain instances are more likely to be mislabeled based on the characteristics of their features. For instance-dependent noise two types are commonly used:

**IDN-C** Chen et al. (2021) utilizes deep neural network trained on clean data to simulate the label noise. For each training instance $(\mathbf{x}_i, y_i)$, the softmax outputs $f^{(e_i)}(\mathbf{x}_i)$ are recorded at each epoch, and the average predicted probability distribution is computed as: $\mathbf{p}_i = \frac{1}{e_t} \sum_{t=1}^{e_t} f^{(t)}(\mathbf{x}_i)$, where $e_t$ are the total epochs. The most confusing incorrect class is identified as the one having maximum probability in $\mathbf{p}_i$ defined as its confidence score $s_i = \mathbf{p}_i[\tilde{y}_i]$. The top $\eta n$ instances with the highest confidence scores are selected, and their labels are flipped to $\tilde{y}_i$, creating instance-dependent noise at a specified noise rate $\eta$.

**Badlabel** Zhang et al. (2024b) is an adversarial approach that generates noisy labels by selecting a subset of instances where the loss values for clean and noisy labels are nearly indistinguishable. A model is first trained on the clean dataset $\mathcal{D}_{\text{clean}}$ to compute the loss $\ell(\mathbf{x}_i, y_i)$ for each instance. A subset $\mathcal{S} \subset \mathcal{D}_{\text{clean}}$ of cardinality $\eta n$, is selected by identifying instances that produce loss values with incorrect labels similar to that of the correct labels. More specifically, for each instance $(\mathbf{x}_i, y_i) \in \mathcal{S}$, the label $y_i$ is changed to an incorrect label $\tilde{y}_i \neq y_i$ such that the loss $\ell(\mathbf{x}_i, \tilde{y}_i)$ is close to $\ell(\mathbf{x}_i, y_i)$, maximizing confusion for the model.

# B ADDITIONAL DETAILS

## B.1 DATASETS

We evaluated the performance across eight datasets: CIFAR-10, CIFAR-100, CIFAR80N-O Wan et al. (2024), CIFAR10N, CIFAR100N, WebVision Li et al. (2017a), Clothing1M Xiao et al. (2015), and Food-101N Lee et al. (2018).

**CIFAR-10 & CIFAR-100** consists of 60,000 color images of size $32 \times 32$ pixels, categorized into 10 and 100 classes, with 6,000 and 600 images per class, respectively. We followed the standard 50,000/10,000 train/test split and report test results for both CIFAR-10 and CIFAR-100. Additionally, we reserved 10% of the training data as the validation set for CIFAR-10 and CIFAR-100.

**CIFAR-80N** is an open-set noisy label dataset derived from CIFAR-100, specifically curated to study the effects of open-set noise in supervised learning. The dataset is constructed by selecting 80 classes from the original CIFAR-100 dataset as the in-distribution (ID) or clean classes, and treating the remaining 20 classes as out-of-distribution (OOD) noise sources. To simulate open-set label noise, the training samples from the 20 OOD classes are randomly placed within the 80 ID classes by randomly assigning them labels from the 80 ID classes. This mimics real-world scenarios where mislabeled data originates from unknown or unseen categories. The test set remains clean and contains only correctly labeled samples from the 80 in-distribution classes. CIFAR-80N preserves the image resolution and format of CIFAR-100 ($32 \times 32$ color images), and follows the same 50,000/10,000 train/test split, with 10% of the training set reserved for validation.

**CIFAR-10N & CIFAR-100N** Wei et al. (2022) are noisy versions of the standard CIFAR-10 and CIFAR-100 datasets, designed to reflect real-world human annotation errors. Each sample in CIFAR-10N and CIFAR-100N retains the original image from CIFAR-10/100 but is annotated with

a noisy label provided by multiple human annotators via Amazon Mechanical Turk. For CIFAR-10N, five distinct noisy label variants are provided: *aggre* (aggregated labels via majority vote), *random1*, *random2*, *random3* (three independent annotator responses), and *worst* (labels selected from the most error-prone annotators). The estimated noise rates are approximately: 9.03% for *aggre*, 18% for the *random* variants, and 40.21% for *worst*. CIFAR-100N follows a similar setup with a single noisy annotation per image and an estimated average label noise rate of 40.2%. Both datasets use the standard CIFAR training/test splits (50,000/10,000), and we follow the same setup, reserving 10% of the training data as a validation set.

**WebVision** dataset contains over 2.4 million images collected from the web using search queries based on the 1,000 classes of the ILSVRC 2012 benchmark Deng et al. (2009). For our experiments, we used the "mini" version of WebVision, as proposed by Ye et al. (2023); Jiang et al. (2018), focusing on the first 50 classes from the Google resized image subset.

**Clothing1M** Xiao et al. (2015) is a dataset of images of clothing items collected from online retail websites, divided into 14 classes. It contains one million images with noisy labels, primarily due to automated annotations derived from the surrounding text.

**Food-101N** Lee et al. (2018) consists of 310,009 images across 101 food categories. The training set contains 307,147 images with real-world noisy labels collected from web sources, while the test set includes 2,962 clean images.

### B.2 IMPLEMENTATION DETAILS

We evaluate two Vision Transformer backbones (ViT-B/16 and ViT-L/16), both pre-trained on ImageNet-21k Dosovitskiy & et al. (2020). Following the optimization strategy of Ye et al. Ye et al. (2023), we use an SGD optimizer 0.90 momentum and a weight decay of $1 \times 10^{-4}$ for CIFAR-10 and CIFAR10N, and $1 \times 10^{-5}$ for CIFAR-100, CIFAR100N and CIFAR80N-O. For WebVision, Clothing1M, and Food-101N, we use Nesterov momentum of 0.90 and weight decay of $3 \times 10^{-5}$ were used. The initial learning rate is set uniformly at 0.001, with a batch size of 256 and gradient norm clipping at 5.0 across all setups Ye et al. (2023). Baseline method hyperparameters are consistent with those used in the original papers. We will release the code along with all training settings upon acceptance.

## C ViT-B/16 BACKBONE ADDITIONAL RESULTS

### C.1 RQ1 RESULTS

Tables 5, 7, 8, and 9, provide a detailed breakdown of fine-tuning technique performance across various label noise settings. Specifically, Table 7 reports results on clean data and six closed-set noise variants for CIFAR-10 and CIFAR-100. Table 8 presents results for the open-set noise setting using the CIFAR80N-O dataset. Tables 9 and 5 show performance under real-world label noise using the CIFAR-10N and CIFAR-100N datasets, respectively. These tables serve as a detailed extension of the summary results presented in Table 1 of the main paper.

Table 5: **RQ1:** Performance breakdown for real-world label noise across fine-tuning techniques on CIFAR-100N.

| Fine-tuning Techniques | Clean | Noisy100 |
|---|---|---|
| Full-FT | 91.96 | 70.62 |
| AdaptFormer | 83.82 | 70.57 |
| VPT | 89.32 | 70.57 |
| MLP-K | 86.12 | 68.75 |
| LP | 86.12 | 62.5 |

### C.2 RQ2 RESULTS

Table 10 presents a detailed performance comparison between CNN and ViT-B/16 architectures across four closed-set noise subcategories: symmetric, asymmetric, IDN-C, and BadLabel. On

Table 6: **RQ1:** Performance breakdown for real-world label noise across fine-tuning techniques on CIFAR-100N using ViT-L/16.

| Fine-tuning Techniques | Clean | Noisy100 |
|---|---|---|
| Full-FT | 92.68 | 68.73 |
| AdaptFormer | 85.32 | 71.19 |
| VPT | 91.12 | 69.05 |
| MLP-K | 88.4 | 69.53 |
| LP | 85.8 | 62.89 |

Table 7: **RQ1:** Performance breakdown for closed-set label noise across various fine-tuning techniques on CIFAR-10 and CIFAR-100.

| Datasets | Fine-tuning Techniques | Clean | Symmetric noise rate ($\eta$) | | | Asymmetric noise rate ($\eta$) | | |
|---|---|---|---|---|---|---|---|---|
| | | | 0.4 | 0.6 | 0.8 | 0.2 | 0.3 | 0.4 |
| CIFAR-10 | Full-FT | 98.91 | 33.26 | 28.37 | 26.15 | 89.38 | 44.24 | 40.89 |
| | AdaptFormer | 97.27 | 86.89 | 67.32 | 47.31 | 92.75 | 81.05 | 60.18 |
| | VPT | 97.90 | 91.67 | 63.12 | 33.11 | 92.43 | 87.64 | 72.28 |
| | MLP-K | 96.80 | 86.94 | 66.66 | 42.03 | 93.98 | 90.57 | 85.61 |
| | LP | 96.55 | 95.08 | 92.21 | 62.5 | 88.6 | 76.04 | 61.58 |
| CIFAR-100 | Full-FT | 91.96 | 50.32 | 28.12 | 25.48 | 49.38 | 34.24 | 30.89 |
| | AdaptFormer | 83.82 | 67.61 | 59.76 | 52.16 | 65.75 | 57.05 | 45.18 |
| | VPT | 89.32 | 67.45 | 50.35 | 25.54 | 68.43 | 52.64 | 39.28 |
| | MLP-K | 86.12 | 58.98 | 44.66 | 38.71 | 70.17 | 60.02 | 48.17 |
| | LP | 86.12 | 68.35 | 58.07 | 48.51 | 55.07 | 46.48 | 40.10 |

average, ViT-B/16 demonstrates greater robustness, outperforming CNNs by 10.58% on CIFAR-10 and 11.64% on CIFAR-100. This table provides a more granular view of the averaged results previously summarized in Table 2 of the main paper.

## C.3 RQ3 RESULTS

Tables 11 and 12 report the mean test accuracy and standard deviation over three runs for symmetric, asymmetric, IDN-C, and BadLabel closed-set noise on the CIFAR-10 and CIFAR-100 datasets, using the ViT-B/16 backbone fine-tuned with MLP-K. These tables provide detailed results supporting Figure 3 in the main paper. Similarly, Tables 13 and 19 present the results for open-set and real-world noisy label datasets for ViT-B/16 with MLP-K fine-tuning, extending the Figure 4 of the main paper.

## C.4 RQ4 RESULTS

Table 20 reports the test accuracy and corresponding prediction entropy reduction for twelve benchmarked methods across three noise categories: closed-set, open-set, and real-world label noise. This table presents the data used to generate Figure 5 in the main paper. To provide a more granular analysis, Table 21 details the entropy reduction results for closed-set noisy settings on the CIFAR-10 and CIFAR-100 datasets. Similarly, Table 14 shows the entropy reduction for the CIFAR80N-O dataset under open-set noise. Lastly, Table 22 presents the entropy reduction across all twelve methods on five real-world noisy datasets.

Table 8: **RQ1:** Performance breakdown for open-set label noise across various fine-tuning techniques on CIFAR80N-O.

| Datasets | Fine-tuning Techniques | Clean | Symmetric noise rate ($\eta$) | | Asymmetric noise rate ($\eta$) |
|---|---|---|---|---|---|
| | | | 0.2 | 0.8 | 0.4 |
| CIFAR80N-O | Full-FT | 90.68 | 55.05 | 28.96 | 38.24 |
| | AdaptFormer | 92.12 | 75.89 | 35.63 | 53.5 |
| | VPT | 91.28 | 74.24 | 33.72 | 55.45 |
| | MLP-K | 89.72 | 85.15 | 37.89 | 71.87 |
| | LP | 89.45 | 83.52 | 34.14 | 70.72 |

Table 9: **RQ1:** Performance breakdown for real-world label noise across fine-tuning techniques on CIFAR-10N.

| Datasets | Fine-tuning Techniques | Clean | Noise Subset | | | | |
|---|---|---|---|---|---|---|---|
| | | | Aggregate | Random1 | Random2 | Random3 | Worst |
| CIFAR-10N | Full-FT | 98.91 | 95.46 | 35.18 | 34.95 | 34.97 | 29.63 |
| | AdaptFormer | 97.27 | 96.15 | 88.72 | 88.36 | 87.89 | 72.18 |
| | VPT | 97.90 | 96.54 | 92.78 | 92.46 | 92.53 | 69.72 |
| | MLP-K | 96.80 | 94.53 | 95.31 | 94.53 | 93.75 | 90.23 |
| | LP | 96.55 | 94.14 | 93.75 | 94.14 | 93.75 | 89.84 |

Table 10: **RQ2:** Performance of CNN and ViT-B/16 backbones on CIFAR-10 and CIFAR-100 under closed-set label noise.

| Datasets | Arch. | Symmetric | | | Asymmetric | | | IDN-C | | BadLabel | | Avg. |
|---|---|---|---|---|---|---|---|---|---|---|---|---|
| | | 0.4 | 0.6 | 0.8 | 0.2 | 0.3 | 0.4 | 0.6 | 0.8 | 0.6 | 0.8 | |
| **CIFAR-10** | CNN | 58.19 | 38.75 | 19.09 | 83 | 78.15 | 73.69 | 52.22 | 28.04 | 35.66 | 13.44 | 48.02 |
| | ViT-B/16 | 86.94 | 66.66 | 32.03 | 93.98 | 90.57 | 84.61 | 48.05 | 22.26 | 44.14 | 16.79 | 58.60 |
| **CIFAR-100** | CNN | 40.72 | 22.98 | 7.55 | 58.25 | 50.3 | 41.53 | 52.55 | 40.45 | 17.05 | 4.18 | 33.56 |
| | ViT-B/16 | 58.98 | 41.66 | 36.71 | 70.17 | 60.02 | 48.17 | 58.98 | 37.5 | 31.25 | 8.59 | 45.20 |

Table 11: **RQ3:** Test Accuracy (mean±std) comparison for SLF and NLL methods on CIFAR-10 and CIFAR-100 datasets for ViT-B/16 backbone using MLP-K finetuning. Performance is reported for symmetric and asymmetric closed-set label noise for noise rate $\eta_{sym} \in \{0.4, 0.6, 0.8\}$ and $\eta_{asym} \in \{0.2, 0.3, 0.4\}$.

| | Method | Clean | Symmetric Noise Rate ($\eta$) | | | Asymmetric Noise Rate ($\eta$) | | |
|---|---|---|---|---|---|---|---|---|
| | | | 0.4 | 0.6 | 0.8 | 0.2 | 0.3 | 0.4 |
| CIFAR-10 | CE | 96.80±0.04 | 86.94±0.31 | 66.66±0.15 | 32.03±0.44 | 93.98±0.03 | 90.57±0.18 | 84.61±0.30 |
| | FL | 96.50±0.07 | 88.64±0.32 | 70.81±0.04 | 33.47±0.41 | 95.27±0.03 | 93.39±0.12 | 88.20±0.12 |
| | GCE | 96.40±0.03 | 6.16±0.04 | 95.63±0.04 | 92.70±0.06 | 94.15±0.01 | 94.97±0.10 | 88.89±0.42 |
| | SCE | 96.36±0.04 | 94.98±0.02 | 89.58±0.22 | 48.88±1.03 | 95.48±0.09 | 92.40±0.20 | 84.58±0.17 |
| | NLNL | 95.42±0.06 | 85.32±0.02 | 20.03±0.03 | 10.00±0.01 | 86.37±0.17 | 82.05±0.01 | 78.07±0.07 |
| | DivideMix | 96.5±0.32 | 96.25±0.18 | 96.11±0.45 | 95.52±0.25 | 96.16±0.17 | 95.58±0.69 | 94.73±0.48 |
| | APL | 96.28±0.05 | 95.96±0.05 | 95.12±0.13 | 89.66±0.07 | 96.20±0.10 | 95.66±0.07 | 75.19±0.59 |
| | NCE+AGCE | 96.31±0.03 | 95.81±0.08 | 94.53±0.07 | 88.90±0.58 | 94.53±0.12 | 84.37±0.09 | 67.57±1.06 |
| | ANL | 95.83±0.18 | 94.92±0.63 | 94.27±0.48 | 76.17±0.16 | 96.61±0.48 | 95.70±0.84 | 94.14±0.31 |
| | Robust DivideMix | 96.92±0.17 | 96.18±0.12 | 96.16±0.24 | 95.81±0.18 | 95.53±0.22 | 95.03±0.35 | 87.66±0.17 |
| | CLIPCleaner | 94.53±0.69 | 91.38±0.30 | 91.01±0.32 | 90.92±0.18 | 91.41±0.12 | 91.25±0.69 | 91.16±0.45 |
| | NoiseGPT | 96.39±0.11 | 96.11±0.45 | 95.52±0.18 | 87.04±0.44 | 96.82±0.15 | 95.97±0.31 | 92.47±0.95 |
| CIFAR-100 | CE | 86.12±0.97 | 58.98±0.55 | 41.66±1.28 | 36.71±1.22 | 70.17±0.12 | 60.02±0.20 | 48.17±1.75 |
| | FL | 83.20±0.55 | 69.80±0.75 | 42.44±0.40 | 22.78±0.66 | 71.34±0.63 | 62.23±0.29 | 52.08±0.10 |
| | GCE | 83.46±0.55 | 82.42±0.80 | 79.29±1.98 | 75.38±1.77 | 82.03±0.73 | 76.55±0.68 | 57.80±0.18 |
| | SCE | 83.20±0.48 | 61.19±0.40 | 47.26±0.92 | 28.51±0.39 | 73.56±0.80 | 60.93±0.14 | 51.55±0.77 |
| | NLNL | 74.33±0.63 | 52.92±0.36 | 38.52±0.11 | 10.41±0.13 | 63.14±0.04 | 41.84±0.13 | 36.59±0.18 |
| | DivideMix | 84.17±0.05 | 82.51±0.13 | 80.57±0.40 | 75.51±0.18 | 82.02±0.34 | 80.96±0.32 | 76.98±0.48 |
| | APL | 84.42±0.76 | 82.42±0.38 | 80.07±0.55 | 77.34±0.14 | 83.20±0.31 | 78.25±0.28 | 64.71±0.73 |
| | NCE+AGCE | 84.11±0.11 | 82.81±0.84 | 81.37±1.02 | 78.25±1.41 | 83.85±0.97 | 81.63±0.10 | 70.83±0.97 |
| | ANL | 83.79±0.70 | 83.20±0.68 | 81.50±1.21 | 65.75±1.57 | 82.55±0.80 | 82.52±0.69 | 77.34±0.95 |
| | Robust DivideMix | 84.08±0.18 | 82.23±0.11 | 81.87±0.23 | 79.28±0.52 | 81.84±0.34 | 80.87±0.69 | 79.29±1.02 |
| | CLIPCleaner | 78.26±0.04 | 68.05±0.18 | 67.90±0.09 | 66.56±0.13 | 68.49±0.24 | 68.20±0.20 | 67.99±0.29 |
| | NoiseGPT | 83.19±0.52 | 78.72±0.32 | 75.39±0.42 | 69.57±0.64 | 80.56±0.32 | 79.87±0.45 | 79.18±0.11 |

Table 12: **RQ3:** Test Accuracy (mean±std) comparison for SLF and NLL methods on CIFAR-10 and CIFAR-100 datasets for ViT-B/16 backbone using MLP-K finetuning. Performance is reported for IDN-C and BadLabel closed-set label noise for noise rate $\eta \in \{0.6, 0.8\}$.

| | Method | Clean | IDN-C Noise Rate ($\eta$) | | BadLabel Noise Rate ($\eta$) | |
|---|---|---|---|---|---|---|
| | | | 0.6 | 0.8 | 0.6 | 0.8 |
| CIFAR-10 | CE | 96.80±0.04 | 48.05±0.12 | 22.26±0.18 | 44.14±0.13 | 16.79±0.95 |
| | FL | 96.50±0.07 | 50.00±0.15 | 17.59±0.12 | 40.23±0.42 | 18.35±0.30 |
| | GCE | 96.40±0.03 | 55.08±0.02 | 19.53±0.06 | 41.41±0.41 | 15.62±0.18 |
| | SCE | 96.36±0.04 | 51.56±0.40 | 19.14±0.12 | 44.14±0.63 | 20.70±0.75 |
| | NLNL | 95.42±0.06 | 75.78±0.68 | 45.31±0.84 | 41.01±0.11 | 9.76±0.32 |
| | DivideMix | 96.5±0.32 | 68.29±0.09 | 26.70±0.18 | 37.63±0.52 | 11.25±0.69 |
| | APL | 96.28±0.05 | 54.69±0.55 | 19.14±0.23 | 49.22±0.25 | 15.23±0.44 |
| | NCE+AGCE | 96.31±0.03 | 63.67±0.12 | 17.58±0.48 | 43.75±0.95 | 14.06±0.77 |
| | ANL | 95.83±0.18 | 58.59±0.12 | 21.09±0.19 | 43.35±0.55 | 12.89±0.63 |
| | Robust DivideMix | 96.92±0.17 | 69.30±0.73 | 27.19±0.64 | 36.70±0.13 | 13.40±0.52 |
| | CLIPCleaner | 94.53±0.69 | 90.62±0.59 | 79.35±0.48 | 91.52±0.95 | 80.92±0.73 |
| | NoiseGPT | 96.39±0.11 | 94.15±0.66 | 79.02±0.77 | 93.12±0.10 | 84.72±0.17 |
| CIFAR-100 | CE | 86.12±0.97 | 58.98±0.11 | 37.5±0.09 | 31.25±0.22 | 8.59±0.34 |
| | FL | 83.20±0.55 | 59.76±0.32 | 39.84±0.24 | 34.37±0.53 | 7.31±0.11 |
| | GCE | 83.46±0.55 | 57.81±0.12 | 36.32±0.24 | 24.21±0.18 | 3.12±0.19 |
| | SCE | 83.20±0.48 | 63.67±0.12 | 41.79±0.06 | 19.92±0.09 | 1.17±0.19 |
| | NLNL | 74.33±0.63 | 49.22±0.15 | 44.14±0.13 | 14.06±0.18 | 5.47±0.21 |
| | DivideMix | 84.17±0.05 | 68.78±0.24 | 51.22±0.28 | 48.96±0.38 | 15.59±0.32 |
| | APL | 84.42±0.76 | 64.68±0.62 | 45.31±0.73 | 34.37±0.85 | 3.51±0.90 |
| | NCE+AGCE | 84.11±0.11 | 64.84±0.44 | 48.05±0.12 | 30.07±0.17 | 1.95±0.19 |
| | ANL | 83.79±0.70 | 65.62±0.48 | 46.09±0.42 | 34.37±0.58 | 9.37±0.73 |
| | Robust DivideMix | 84.08±0.18 | 69.97±0.43 | 51.69±0.12 | 47.51±0.18 | 13.96±0.19 |
| | CLIPCleaner | 78.26±0.04 | 65.17±0.18 | 61.42±0.95 | 50.80±0.13 | 23.50±0.28 |
| | NoiseGPT | 83.19±0.52 | 66.46±0.44 | 62.39±0.68 | 54.72±0.32 | 25.32±0.24 |

Table 13: **RQ3:** Test Accuracy (mean±std) comparison for SLF and NLL methods for open-set label noise on CIFAR80N-O dataset for ViT-B/16 backbone using MLP-K finetuning.

| **Method** | **Sym Noise ($\eta$)** | | **Asym Noise ($\eta$)** |
|---|---|---|---|
| | 0.2 | 0.8 | 0.4 |
| CE | 85.15±0.12 | 37.89±0.18 | 71.87±0.32 |
| FL | 82.03±0.08 | 39.06±0.48 | 67.57±0.73 |
| GCE | 87.89±0.18 | 80.85±0.11 | 66.79±0.21 |
| SCE | 82.42±0.17 | 46.87±0.24 | 63.67±0.05 |
| NLNL | 46.48±0.35 | 37.5±0.48 | 59.76±0.26 |
| DivideMix | 84.21±0.13 | 81.06±0.18 | 72.56±0.32 |
| APL | 87.89±0.24 | 80.85±0.52 | 77.73±0.62 |
| NCE+AGCE | 87.5±0.48 | 85.93±0.53 | 79.68±0.30 |
| ANL | 89.45±0.44 | 83.2±0.52 | 77.73±0.11 |
| Robust DivideMix | 84.03±0.07 | 81.5±0.11 | 70.76±0.05 |
| CLIPCleaner | 72.31±0.07 | 70.73±0.12 | 71.97±0.15 |
| NoiseGPT | 73.11±0.18 | 72.24±0.24 | 72.56±0.11 |

Table 14: **RQ4:** Prediction Entropy Reduction breakdown for CIFAR80N-O across two standard classification losses (SLF) and ten SOTA NLL methods using the ViT-B/16 backbone for open-set noise settings.

| Method | Sym Noise ($\eta$) | | Asym Noise ($\eta$) |
|---|---|---|---|
| | 0.2 | 0.8 | 0.4 |
| CE | 0.485 | 0.266 | 0.561 |
| FL | 0.452 | 0.257 | 0.557 |
| GCE | 0.928 | 0.868 | 0.888 |
| SCE | 0.559 | 0.281 | 0.659 |
| NLNL | 0.232 | 0.012 | 0.474 |
| DivideMix | 0.513 | 0.516 | 0.428 |
| APL | 0.95 | 0.904 | 0.933 |
| NCE+AGCE | 0.949 | 0.905 | 0.939 |
| ANL | 0.991 | 0.98 | 0.987 |
| Robust DivideMix | 0.51 | 0.47 | 0.413 |
| CLIPCleaner | 0.799 | 0.736 | 0.74 |
| NoiseGPT | 0.769 | 0.746 | 0.752 |

Table 15: **RQ5:** Impact of entropy regularization on performance for CIFAR80N-O under open-set noisy settings using ViT-B/16 backbone.

| Method | Sym Noise ($\eta$) | | Asym Noise ($\eta$) |
|---|---|---|---|
| | 0.2 | 0.8 | 0.4 |
| CE | 85.15 | 37.89 | 71.87 |
| CE+$H$ | 88.28 | 75 | 72.26 |
| | ↑3.13 | ↑37.11 | ↑0.39 |
| FL | 82.03 | 39.06 | 67.57 |
| FL+$H$ | 87.89 | 77.34 | 73.04 |
| | ↑5.86 | ↑38.28 | ↑5.47 |

Table 16: **RQ3:** Test Accuracy (mean±std) comparison for SLF and NLL methods for open-set label noise on CIFAR80N-O dataset for ViT-L/16 backbone using MLP-K finetuning.

| Method | Sym Noise ($\eta$) | | Asym Noise ($\eta$) |
|---|---|---|---|
| | 0.2 | 0.8 | 0.4 |
| CE | 78.51±0.24 | 26.17±0.11 | 55.85±0.24 |
| FL | 81.64±0.10 | 27.34±0.23 | 58.2±0.53 |
| GCE | 87.11±0.15 | 76.95±0.19 | 59.76±0.44 |
| SCE | 82.42±0.32 | 28.12±0.45 | 57.81±0.09 |
| NLNL | 84.37±0.31 | 78.12±0.12 | 57.03±0.42 |
| DivideMix | 87.59±0.12 | 84.62±0.52 | 63.41±0.63 |
| APL | 87.1±0.85 | 75.39±0.44 | 62.5±0.38 |
| NCE+AGCE | 86.71±0.63 | 76.56±1.01 | 67.57±0.19 |
| ANL | 88.67±0.52 | 82.42±0.89 | 67.96±0.95 |
| Robust DivideMix | 87.3±0.18 | 84.81±0.15 | 66.36±0.29 |
| CLIPCleaner | 82.42±0.38 | 80.25±0.28 | 81.46±0.63 |
| NoiseGPT | 82.84±0.72 | 81.32±0.85 | 82.09±0.99 |

Table 17: **RQ4:** Prediction Entropy Reduction breakdown for CIFAR80N-O across two standard classification losses (SLF) and ten SOTA NLL methods using the ViT-L/16 backbone for open-set noise settings.

| Method | Sym Noise ($\eta$) | | Asym Noise ($\eta$) |
|---|---|---|---|
| | 0.2 | 0.8 | 0.4 |
| CE | 0.63 | 0.467 | 0.571 |
| FL | 0.84 | 0.347 | 0.681 |
| GCE | 0.96 | 0.898 | 0.757 |
| SCE | 0.882 | 0.393 | 0.68 |
| NLNL | 0.596 | 0.738 | 0.685 |
| DivideMix | 0.861 | 0.855 | 0.759 |
| APL | 0.959 | 0.815 | 0.757 |
| NCE+AGCE | 0.958 | 0.816 | 0.958 |
| ANL | 0.993 | 0.985 | 0.993 |
| Robust DivideMix | 0.747 | 0.735 | 0.607 |
| CLIPCleaner | 0.841 | 0.825 | 0.839 |
| NoiseGPT | 0.781 | 0.74 | 0.719 |

Table 18: **RQ5:** Impact of entropy regularization on performance for CIFAR80N-O under open-set noisy settings using ViT-L/16 backbone.

| Method | Sym Noise ($\eta$) | | Asym Noise ($\eta$) |
|---|---|---|---|
| | 0.2 | 0.8 | 0.4 |
| CE | 78.51 | 26.17 | 55.85 |
| CE+$H$ | 87.89 | 79.29 | 66.41 |
| | ↑9.38 | ↑53.12 | ↑10.56 |
| FL | 81.64 | 27.34 | 58.2 |
| FL+$H$ | 87.5 | 77.73 | 66.02 |
| | ↑5.86 | ↑50.39 | ↑7.82 |

Table 19: **RQ3:** Test Accuracy (mean±std) comparison for SLF and NLL methods on five real-world noisy labels datasets for ViT-B/16 backbone using MLP-K finetuning.

| Method | CIFAR10N | | | | CIFAR100N | WebVision | Clothing1M | Food101N |
|---|---|---|---|---|---|---|---|---|
| | Aggregate | Random1 | Random2 | Random3 | Worst | | | | |
| CE | 94.53±0.09 | 95.31±0.11 | 94.53±0.15 | 93.75±0.09 | 90.23±0.18 | 68.75±0.24 | 88.47±0.21 | 64.94±0.32 | 78.31±0.35 |
| FL | 94.53±0.44 | 95.31±0.32 | 94.14±0.28 | 94.14±0.19 | 90.23±0.31 | 67.58±0.73 | 88.67±0.66 | 62.21±1.21 | 74.71±0.84 |
| GCE | 95.31±0.34 | 94.14±0.10 | 94.53±0.09 | 94.53±0.55 | 91.41±0.97 | 67.58±0.70 | 76.75±0.18 | 65.42±0.04 | 72.16±0.52 |
| SCE | 94.92±0.11 | 95.31±0.15 | 93.75±0.31 | 94.14±0.72 | 89.45±0.63 | 73.05±0.44 | 87.4±0.32 | 62.21±0.18 | 72.94±0.85 |
| NLNL | 94.53±0.63 | 94.14±0.05 | 93.75±0.09 | 94.14±0.18 | 91.01±0.15 | 41.79±0.24 | 12.89±0.45 | 65.91±0.32 | 8.87±0.62 |
| DivideMix | 95.45±0.52 | 95.53±0.22 | 95.53±0.13 | 95.53±0.15 | 95.37±0.45 | 17.21±0.44 | 53.28±0.23 | 60.83±0.21 | 35.89±0.72 |
| APL | 94.92±0.12 | 95.7±0.12 | 93.75±0.08 | 94.92±0.12 | 91.01±0.14 | 74.61±0.11 | 88.57±0.25 | 65.52±0.22 | 74.9±0.44 |
| NCE+AGCE | 95.31±0.06 | 94.92±0.08 | 94.53±0.11 | 95.31±0.25 | 91.01±0.41 | 71.09±0.32 | 89.35±0.63 | 64.64±0.72 | 75.18±0.45 |
| ANL | 94.92±0.21 | 95.31±0.22 | 94.14±0.26 | 94.53±0.44 | 91.79±0.24 | 74.6±0.07 | 89.16±0.75 | 64.35±1.08 | 72.55±0.95 |
| Robust DivideMix | 93.94±0.11 | 94.92±0.24 | 95.27±0.25 | 95.15±0.18 | 94.21±0.32 | 76.99±0.42 | 81.84±0.34 | 61.9±0.45 | 70.87±1.01 |
| CLIPCleaner | 91.42±0.09 | 91.63±1.21 | 91.46±0.95 | 91.58±0.99 | 91.26±0.85 | 67.08±0.73 | 82.14±0.62 | 68.1±0.75 | 71.18±0.54 |
| NoiseGPT | 95.49±0.19 | 95.48±0.12 | 95.41±0.16 | 95.39±0.17 | 96.17±0.23 | 80.1±0.25 | 83.82±0.36 | 70.24±0.32 | 71.68±0.29 |

Table 20: **RQ4:** Entropy Reduction and Robustness (Accuracy) across two standard classification losses and ten SOTA NLL methods using the ViT-B/16 backbone.

| Method | Closed-Set Noise | | Open-Set Noise | | Real-Wrold Noise | |
|---|---|---|---|---|---|---|
| | Acc. | Entropy Reduction | Acc. | Entropy Reduction | Acc. | Entropy Reduction |
| CE | 51.903 | 0.483 | 64.97 | 0.437 | 85.424 | 0.663 |
| FL | 52.895 | 0.465 | 62.887 | 0.422 | 84.613 | 0.608 |
| GCE | 63.454 | 0.861 | 78.51 | 0.895 | 83.537 | 0.874 |
| SCE | 54.55 | 0.675 | 64.32 | 0.5 | 84.797 | 0.729 |
| NLNL | 47.87 | 0.427 | 47.913 | 0.239 | 66.337 | 0.435 |
| DivideMix | 69.066 | 0.748 | 79.277 | 0.486 | 71.624 | 0.522 |
| APL | 64.997 | 0.913 | 82.157 | 0.929 | 85.989 | 0.856 |
| NCE+AGCE | 64.421 | 0.895 | 84.37 | 0.931 | 85.704 | 0.859 |
| ANL | 65.802 | 0.94 | 83.46 | 0.986 | 85.707 | 0.889 |
| Robust DivideMix | 69.074 | 0.764 | 78.763 | 0.464 | 85.01 | 0.637 |
| CLIPCleaner | 74.881 | 0.795 | 71.67 | 0.758 | 82.872 | 0.814 |
| NoiseGPT | 79.356 | 0.84 | 72.637 | 0.756 | 87.087 | 0.855 |

Table 21: **RQ4:** Prediction Entropy Reduction breakdown for CIFAR-10 and CIFAR-100 across two standard classification losses (SLF) and ten SOTA NLL methods using the ViT-B/16 backbone for closed-set noise settings.

| | Method | Symmetric Noise ($\eta$) | | | Asymmetric Noise ($\eta$) | | | IDN-C Noise ($\eta$) | | BadLabel Noise ($\eta$) | |
|---|---|---|---|---|---|---|---|---|---|---|---|
| | | 0.4 | 0.6 | 0.8 | 0.2 | 0.3 | 0.4 | 0.6 | 0.8 | 0.6 | 0.8 |
| CIFAR-10 | CE | 0.272 | 0.185 | 0.097 | 0.666 | 0.635 | 0.613 | 0.525 | 0.504 | 0.56 | 0.532 |
| | FL | 0.256 | 0.174 | 0.09 | 0.652 | 0.622 | 0.609 | 0.505 | 0.48 | 0.535 | 0.509 |
| | GCE | 0.921 | 0.855 | 0.263 | 0.927 | 0.878 | 0.842 | 0.811 | 0.792 | 0.827 | 0.782 |
| | SCE | 0.844 | 0.784 | 0.657 | 0.904 | 0.874 | 0.818 | 0.776 | 0.745 | 0.768 | 0.694 |
| | NLNL | 0.652 | 0.512 | 0.292 | 0.667 | 0.647 | 0.652 | 0.535 | 0.423 | 0.358 | 0.295 |
| | DivideMix | 0.891 | 0.886 | 0.875 | 0.893 | 0.873 | 0.859 | 0.743 | 0.632 | 0.691 | 0.43 |
| | APL | 0.954 | 0.96 | 0.944 | 0.939 | 0.923 | 0.87 | 0.866 | 0.782 | 0.922 | 0.698 |
| | NCE+AGCE | 0.957 | 0.957 | 0.948 | 0.946 | 0.914 | 0.787 | 0.828 | 0.79 | 0.779 | 0.608 |
| | ANL | 0.991 | 0.989 | 0.966 | 0.991 | 0.989 | 0.983 | 0.951 | 0.853 | 0.882 | 0.824 |
| | Robust DivideMix | 0.981 | 0.982 | 0.973 | 0.975 | 0.969 | 0.943 | 0.72 | 0.551 | 0.531 | 0.502 |
| | CLIPCleaner | 0.937 | 0.931 | 0.917 | 0.927 | 0.915 | 0.913 | 0.919 | 0.858 | 0.942 | 0.887 |
| | NoiseGPT | 0.973 | 0.964 | 0.934 | 0.961 | 0.937 | 0.922 | 0.935 | 0.84 | 0.948 | 0.862 |
| CIFAR-100 | CE | 0.381 | 0.296 | 0.247 | 0.653 | 0.625 | 0.612 | 0.643 | 0.578 | 0.532 | 0.512 |
| | FL | 0.347 | 0.284 | 0.228 | 0.626 | 0.615 | 0.611 | 0.604 | 0.545 | 0.529 | 0.486 |
| | GCE | 0.947 | 0.93 | 0.887 | 0.955 | 0.947 | 0.926 | 0.955 | 0.943 | 0.925 | 0.905 |
| | SCE | 0.439 | 0.343 | 0.259 | 0.73 | 0.706 | 0.699 | 0.707 | 0.643 | 0.615 | 0.493 |
| | NLNL | 0.435 | 0.271 | 0.124 | 0.576 | 0.487 | 0.435 | 0.496 | 0.429 | 0.158 | 0.092 |
| | DivideMix | 0.827 | 0.814 | 0.785 | 0.817 | 0.807 | 0.786 | 0.682 | 0.643 | 0.536 | 0.483 |
| | APL | 0.953 | 0.94 | 0.911 | 0.961 | 0.953 | 0.942 | 0.962 | 0.948 | 0.929 | 0.903 |
| | NCE+AGCE | 0.95 | 0.938 | 0.903 | 0.957 | 0.952 | 0.942 | 0.962 | 0.951 | 0.92 | 0.902 |
| | ANL | 0.99 | 0.986 | 0.969 | 0.991 | 0.99 | 0.987 | 0.961 | 0.889 | 0.826 | 0.788 |
| | Robust DivideMix | 0.854 | 0.845 | 0.824 | 0.847 | 0.843 | 0.806 | 0.741 | 0.625 | 0.436 | 0.328 |
| | CLIPCleaner | 0.779 | 0.728 | 0.695 | 0.792 | 0.778 | 0.616 | 0.722 | 0.631 | 0.577 | 0.432 |
| | NoiseGPT | 0.746 | 0.737 | 0.708 | 0.945 | 0.923 | 0.916 | 0.772 | 0.731 | 0.682 | 0.364 |

Table 22: **RQ4:** Prediction Entropy Reduction breakdown for five real-world noisy label datasets across two standard classification losses (SLF) and ten SOTA NLL methods using the ViT-B/16 backbone.

| Method | CIFAR10N | | | | | CIFAR100N | WebVision | Clothing1M | Food101N |
|---|---|---|---|---|---|---|---|---|---|
| | Agg. | Rand1 | Rand2 | Rand3 | Worst | | | | |
| CE | 0.674 | 0.696 | 0.68 | 0.681 | 0.614 | 0.654 | 0.741 | 0.564 | 0.659 |
| FL | 0.645 | 0.665 | 0.653 | 0.659 | 0.594 | 0.543 | 0.713 | 0.439 | 0.563 |
| GCE | 0.978 | 0.976 | 0.976 | 0.974 | 0.924 | 0.963 | 0.916 | 0.333 | 0.823 |
| SCE | 0.95 | 0.925 | 0.927 | 0.922 | 0.827 | 0.648 | 0.609 | 0.41 | 0.339 |
| NLNL | 0.776 | 0.651 | 0.625 | 0.632 | 0.378 | 0.248 | 0.017 | 0.584 | 0.004 |
| DivideMix | 0.633 | 0.634 | 0.636 | 0.633 | 0.563 | 0.364 | 0.338 | 0.522 | 0.377 |
| APL | 0.984 | 0.984 | 0.983 | 0.984 | 0.961 | 0.971 | 0.874 | 0.216 | 0.751 |
| NCE+AGCE | 0.985 | 0.984 | 0.985 | 0.984 | 0.964 | 0.97 | 0.886 | 0.192 | 0.779 |
| ANL | 0.919 | 0.917 | 0.918 | 0.917 | 0.894 | 0.913 | 0.983 | 0.635 | 0.907 |
| Robust DivideMix | 0.625 | 0.645 | 0.648 | 0.645 | 0.634 | 0.656 | 0.696 | 0.503 | 0.68 |
| CLIPCleaner | 0.927 | 0.913 | 0.927 | 0.925 | 0.933 | 0.707 | 0.736 | 0.597 | 0.659 |
| NoiseGPT | 0.963 | 0.962 | 0.957 | 0.956 | 0.954 | 0.867 | 0.753 | 0.615 | 0.667 |

Table 23: **RQ5:** Impact of entropy regularization on performance for CIFAR-10 and CIFAR-100 under closed-set noisy settings using ViT-B/16 backbone.

| Method | | Symmetric Noise ($\eta$) | | | Asymmetric Noise ($\eta$) | | | IDN-C Noise ($\eta$) | | BadLabel Noise ($\eta$) | |
|---|---|---|---|---|---|---|---|---|---|---|---|---|
| | | 0.4 | 0.6 | 0.8 | 0.2 | 0.3 | 0.4 | 0.6 | 0.8 | 0.6 | 0.8 |
| CIFAR-10 | CE | 86.94 | 66.66 | 32.03 | 93.98 | 90.57 | 84.61 | 48.05 | 22.26 | 44.14 | 16.79 |
| | CE+$H$ | 95.87 | 94.35 | 92.96 | 95.12 | 94.66 | 94.18 | 74.22 | 37.69 | 45.31 | 21.87 |
| | | ↑8.93 | ↑27.69 | ↑60.93 | ↑1.14 | ↑4.09 | ↑9.57 | ↑26.17 | ↑15.43 | ↑1.17 | ↑5.08 |
| | FL | 88.64 | 70.81 | 33.47 | 95.27 | 93.39 | 88.2 | 50.00 | 17.59 | 40.23 | 18.35 |
| | FL+$H$ | 94.53 | 90.23 | 57.03 | 97.26 | 95.31 | 93.35 | 70.71 | 33.79 | 46.48 | 21.48 |
| | | ↑5.89 | ↑19.42 | ↑23.56 | ↑1.99 | ↑1.92 | ↑5.15 | ↑20.71 | ↑16.2 | ↑6.25 | ↑3.13 |
| CIFAR-100 | CE | 58.98 | 41.66 | 36.71 | 70.17 | 60.02 | 48.17 | 58.98 | 37.5 | 31.25 | 8.59 |
| | CE+$H$ | 82.68 | 80.33 | 70.04 | 82.89 | 80.33 | 73.43 | 68.94 | 52.93 | 38.67 | 12.89 |
| | | ↑23.7 | ↑38.67 | ↑33.33 | ↑12.72 | ↑20.31 | ↑25.26 | ↑9.96 | ↑15.43 | ↑7.42 | ↑4.3 |
| | FL | 69.80 | 42.44 | 22.78 | 71.34 | 62.23 | 52.08 | 59.76 | 39.84 | 34.37 | 7.31 |
| | FL+$H$ | 84.37 | 78.51 | 66.01 | 81.64 | 79.68 | 73.04 | 66.4 | 49.99 | 39.45 | 10.93 |
| | | ↑14.57 | ↑36.07 | ↑43.23 | ↑10.3 | ↑17.45 | ↑20.96 | ↑6.64 | ↑10.15 | ↑5.08 | ↑3.62 |

Table 24: **RQ5:** Impact of entropy regularization on performance for real-world noisy label datasets using ViT-B/16 backbone.

| Method | CIFAR10N | | | | | CIFAR100N | WebVision | Clothing1M | Food101N |
|---|---|---|---|---|---|---|---|---|---|
| | Agg. | Rand1 | Rand2 | Rand3 | Worst | | | | |
| CE | 94.53 | 95.31 | 94.53 | 93.75 | 90.23 | 68.75 | 88.47 | 64.94 | 78.31 |
| CE+$H$ | 95.31 | 96.09 | 95.31 | 95.31 | 92.97 | 74.22 | 89.35 | 66.53 | 78.83 |
| | ↑0.78 | ↑0.78 | ↑0.78 | ↑1.56 | ↑2.74 | ↑5.47 | ↑0.88 | ↑1.59 | ↑0.52 |
| FL | 94.53 | 95.31 | 94.14 | 94.14 | 90.23 | 67.58 | 88.67 | 62.21 | 74.71 |
| FL+$H$ | 95.31 | 96.09 | 94.92 | 94.92 | 93.36 | 72.26 | 89.16 | 67.48 | 75.1 |
| | ↑0.78 | ↑0.78 | ↑0.78 | ↑0.78 | ↑3.13 | ↑4.68 | ↑0.49 | ↑5.27 | ↑0.39 |

## C.5 RQ5 Results

Tables 23, 15, and 24 present the performance improvements achieved by incorporating entropy regularization into the CE and FL loss functions across closed-set, open-set, and real-world noise categories. Across all settings, entropy regularization consistently enhances the robustness of ViTs to label noise. These tables provide a detailed breakdown of the RQ5 results, extending the findings summarized in Tables 2, 3, and 4.

### C.5.1 The effect of hyperparameter $\lambda$

The effect of the hyperparameter $\lambda$ on performance was evaluated using ViT-B/16 on the CIFAR-10 dataset, with MLP-K fine-tuning. The experiments were categorized into two approaches: 1) keeping $\lambda$ constant at values 0.01, 0.1, 0.2, and 2) linearly increasing $\lambda$ from 0 to 0.3. Figure 6, compares performance across these different $\lambda$ values. A smaller constant $\lambda$ may not fully exploit the benefits of entropy regularization, while higher values could negatively impact the training process. A more effective strategy involves gradually increasing $\lambda$ from 0 to 0.3, resulting in significant performance improvements across different noise levels and fine-tuning techniques. This approach initially prioritizes the baseline loss for learning task-specific features and then gradually shifts focus towards entropy regularization, leading to enhanced robustness in handling noisy labels.

## D ViT-L/16 Backbone Results

In this section, we present the results obtained using the ViT-L/16 backbone, organized by each research question. While our primary analysis in the main paper focuses on the ViT-B/16 model, we extend the evaluation to the larger ViT-L/16 variant to validate the consistency and scalability of our findings across different ViT backbones.

### D.1 RQ1 Results

Tables 25, shows an averaged comparison of five ViT finetuning techniques on clean and noisy datasets across three noise categories. Table 6, 26, 27, and 28, provide a detailed breakdown of

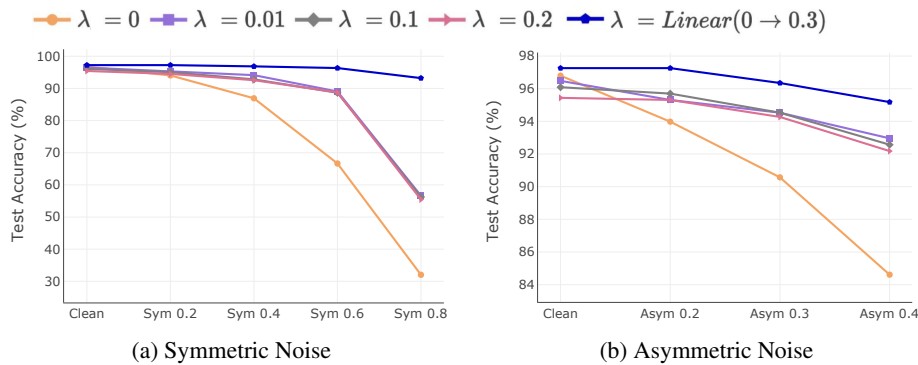

(a) Symmetric Noise        (b) Asymmetric Noise

Figure 6: **Impact of varying $\lambda$ on test accuracy for CIFAR-10 using CE+$\lambda H$ with ViT-B/16+MLP-K under (a) symmetric noise and (b) asymmetric noise.** The linear scheduling of $\lambda$ (Linear(0→0.3)) achieves the best performance across both noise types.

Table 25: **RQ1:** Average performance across clean and noisy settings for closed-set, open-set, and real-world noise using five fine-tuning techniques. Paired t-test (t-values and p-values) reports the significance of degradation.

| Noise | Dataset | Full-FT | AdaptFormer | VPT | MLP-K | LP | t-value | p-value |
|---|---|---|---|---|---|---|---|---|
| **Closed-Set** | Clean | 95.81 | 91.37 | 94.77 | 92.25 | 91.34 | 5.235 | 0.006 |
| | Noisy | 39.63 | 64.46 | 63.18 | 64.44 | 71.33 | | |
| **Open-Set** | Clean | 90.68 | 92.12 | 91.28 | 89.72 | 89.45 | 7.249 | 0.002 |
| | Noisy | 38.3 | 52.99 | 54.28 | 64.97 | 62.79 | | |
| **Real-World** | Clean | 95.81 | 91.37 | 94.77 | 92.25 | 91.34 | 3.762 | 0.019 |
| | Noisy | 56.72 | 78.36 | 77.86 | 79.14 | 78.2 | | |

fine-tuning technique performance across various label noise settings. Specifically, Table 26 reports results on clean data and six closed-set noise variants for CIFAR-10 and CIFAR-100. Table 27 presents results for the open-set noise setting using the CIFAR80N-O dataset. Tables 28 and 6 show performance under real-world label noise using the CIFAR-10N and CIFAR-100N datasets, respectively.

## D.2 RQ2 RESULTS

Table 29, 30, and 31 summarize the results for RQ2 along with t-value and p-value for the three noise categories. Table 32 presents a detailed performance comparison between CNN and ViT-B/16 architectures across four closed-set noise subcategories: symmetric, asymmetric, IDN-C, and BadLabel. On average, ViT-B/16 demonstrates greater robustness, outperforming CNNs by 6.64% on CIFAR-10 and 14.73% on CIFAR-100.

Table 26: **RQ1:** Performance breakdown for closed-set label noise across various fine-tuning techniques on CIFAR-10 and CIFAR-100 using ViT-L/16.

| Datasets | Fine-tuning Techniques | Clean | Symmetric noise rate ($\eta$) | | | Asymmetric noise rate ($\eta$) | | |
|---|---|---|---|---|---|---|---|---|
| | | | 0.4 | 0.6 | 0.8 | 0.2 | 0.3 | 0.4 |
| CIFAR-10 | Full-FT | 98.94 | 30.68 | 29.17 | 24.65 | 87.89 | 43.94 | 39.72 |
| | AdaptFormer | 97.42 | 87.92 | 67.85 | 44.73 | 92.25 | 79.98 | 57.49 |
| | VPT | 98.42 | 90.17 | 64.35 | 35.62 | 91.14 | 88.4 | 73.86 |
| | MLP-K | 96.09 | 81.35 | 57.23 | 25.6 | 93.38 | 88.36 | 79.19 |
| | LP | 96.87 | 94.14 | 89.84 | 58.59 | 93.75 | 92.18 | 83.98 |
| CIFAR-100 | Full-FT | 92.68 | 49.18 | 30.42 | 26.63 | 50.02 | 33.11 | 30.19 |
| | AdaptFormer | 85.32 | 68.46 | 58.63 | 51.07 | 63.55 | 58.11 | 43.5 |
| | VPT | 91.12 | 72.65 | 52.09 | 26.98 | 69.72 | 50.14 | 43.05 |
| | MLP-K | 88.4 | 67.05 | 51.94 | 27.86 | 77.21 | 67.83 | 56.24 |
| | LP | 85.8 | 64.71 | 58.71 | 40.23 | 67.44 | 59.24 | 53.12 |

Table 27: **RQ1:** Performance breakdown for open-set label noise across various fine-tuning techniques on CIFAR80N-O using ViT-L/16.

| Datasets | Fine-tuning Techniques | Clean | Sym noise rate $(\eta)$ | | Asym noise rate $(\eta)$ |
|---|---|---|---|---|---|
| | | | 0.2 | 0.8 | 0.4 |
| CIFAR80N-O | Full-FT | 90.68 | 50.96 | 27.05 | 36.89 |
| | AdaptFormer | 92.12 | 73.14 | 33.74 | 52.09 |
| | VPT | 91.28 | 75.14 | 35.12 | 52.58 |
| | MLP-K | 89.72 | 85.15 | 37.89 | 71.87 |
| | LP | 89.45 | 83.52 | 34.14 | 70.72 |

Table 28: **RQ1:** Performance breakdown for real-world label noise across fine-tuning techniques on CIFAR-10N using ViT-L/16.

| Datasets | Fine-tuning Techniques | Clean | Noise Subset | | | | |
|---|---|---|---|---|---|---|---|
| | | | Aggregate | Random1 | Random2 | Random3 | Worst |
| CIFAR-10N | Full-FT | 98.91 | 95.46 | 35.18 | 34.95 | 34.97 | 29.63 |
| | AdaptFormer | 97.27 | 96.15 | 88.72 | 88.36 | 87.89 | 72.18 |
| | VPT | 97.90 | 96.54 | 92.78 | 92.46 | 92.53 | 69.72 |
| | MLP-K | 96.80 | 94.53 | 95.31 | 94.53 | 93.75 | 90.23 |
| | LP | 96.55 | 94.14 | 93.75 | 94.14 | 93.75 | 89.84 |

Table 29: Comparisons on close-set label noise averaged on CIFAR-10 and CIFAR-100 using ViT-L/16. Standard loss functions (SLF) performance is averaged over cross entropy and Focal loss. Noisy label learning methods (NLL) performance is averaged over GCE, SCE, NLNL, DivideMix, NCE+RCE, NCE+AGCE, ANL-CE, Robust DivideMix, CLIPCleaner, and NoiseGPT. H is the recommended entropy regularization.

| Architecture | Symmetric | | | Asymmetric | | | IDN-C | | BadLabel | | t-value | p-value |
|---|---|---|---|---|---|---|---|---|---|---|---|---|
| | 0.4 | 0.6 | 0.8 | 0.2 | 0.3 | 0.4 | 0.6 | 0.8 | 0.6 | 0.8 | | |
| RQ2: Performance comparison of CNN and ViT with CE. | | | | | | | | | | | | |
| CNN+CE | 49.46 | 30.87 | 13.32 | 70.63 | 64.23 | 57.61 | 52.39 | 34.25 | 26.36 | 8.81 | 3.484 | 0.007 |
| ViT-L/16+CE | 74.2 | 54.59 | 26.73 | 85.29 | 78.09 | 67.72 | 50.86 | 28.83 | 34.37 | 14.06 | | |
| RQ3: Performance comparison of ViTs+SLF and ViTs+NLL. | | | | | | | | | | | | |
| ViT-L/16+SLF | 74.14 | 54.25 | 26.85 | 83.53 | 77.59 | 68.52 | 50.33 | 27.99 | 34.08 | 14.84 | 3.934 | 0.003 |
| ViT-L/16+NLL | 89.34 | 86.09 | 76.87 | 89.58 | 86.46 | 80.13 | 63.88 | 39.51 | 46.79 | 22.91 | | |
| RQ5: Performance comparison of ViT+SLF and ViT+SLF+H, the entropy regularization. | | | | | | | | | | | | |
| ViT-L/16+SLF | 74.14 | 54.25 | 26.85 | 83.53 | 77.59 | 68.52 | 50.32 | 27.98 | 34.08 | 14.84 | 3.404 | 0.008 |
| ViT-L/16+SLF+$H$ | 87.69 | 81.84 | 74.9 | 89.45 | 83.79 | 77.44 | 65.92 | 38.78 | 39.55 | 19.33 | | |

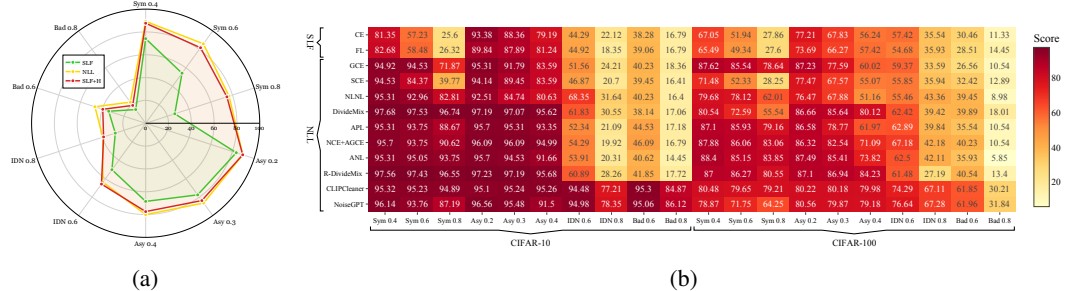

(a)         (b)

Figure 7: **RQ3:** a) Performance comparison of NLL-based methods under closed-set label noise on CIFAR-10 and CIFAR-100 using the ViT-L/16 backbone. b) Detailed results across datasets and noise settings.

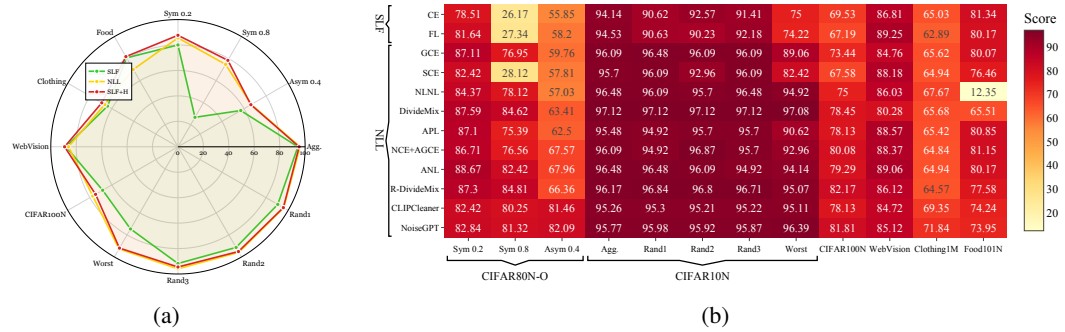

(a)         (b)

Figure 8: **RQ3:** a) Comparison of NLL-based methods using the ViT-L/16 backbone under open-set noise (CIFAR80N-O) and real-world noisy datasets: CIFAR-10N, CIFAR-100N, WebVision, Food101N, and Clothing1M. b) Detailed results across datasets and noise settings.

## D.3  RQ3 RESULTS

Tables 33 and 34 and Figure 7 report the mean test accuracy and standard deviation over three runs for symmetric, asymmetric, IDN-C, and BadLabel closed-set noise on the CIFAR-10 and CIFAR-100 datasets, using the ViT-L/16 backbone fine-tuned with MLP-K. Similarly, Tables 16 and 35 and Figure 8 presents the results for open-set and real-world noisy label datasets for ViT-L/16 with MLP-K fine-tuning.

Table 30: Average performance comparison on CIFAR80N-O under open-set label noise using ViT-L/16. SLF performance is averaged over CE and FL. NLL performance is averaged over ten SOTA methods.

| Architecture | Symmetric | | Asymmetric | t-value | p-value |
|---|---|---|---|---|---|
| | 0.2 | 0.8 | 0.4 | | |
| RQ2: Performance comparison of CNN and ViT with CE. | | | | | |
| CNN+CE | 29.37 | 4.20 | 22.25 | 4.434 | 0.047 |
| ViT-L/16+CE | 78.51 | 26.17 | 55.85 | | |
| RQ3: Performance comparison of ViTs+SLF and ViTs+NLL. | | | | | |
| ViT-L/16+SLF | 80.08 | 26.76 | 57.03 | 1.555 | 0.260 |
| ViT-L/16+NLL | 85.65 | 74.87 | 66.59 | | |
| RQ5: Performance comparison of ViT+SLF and ViT+SLF+H. | | | | | |
| ViT-B/16+SLF | 83.59 | 38.48 | 69.72 | 3.333 | 0.007 |
| ViT-B/16+SLF+$H$ | 88.09 | 76.17 | 72.65 | | |
| ViT-L/16+SLF | 80.08 | 26.76 | 57.03 | | |
| ViT-L/16+SLF+$H$ | 87.70 | 78.51 | 66.22 | | |

Table 31: Average performance comparison on CIFAR10N, CIFAR100N, WebVision, Clothing1M, and Food101N real-world noisy label datasets using ViT-L/16. SLF performance is averaged over CE and FL. NLL performance is averaged over ten SOTA methods.

| Architecture | CIFAR10N | | | | | CIFAR100N | WebVision | Clothing1M | Food101N | t-value | p-value |
|---|---|---|---|---|---|---|---|---|---|---|---|
| | Aggre | Rand1 | Rand2 | Rand3 | Worst | | | | | | |
| RQ2: Performance comparison of CNN and ViT with CE. | | | | | | | | | | | |
| CNN+CE | 87.77 | 85.02 | 86.46 | 85.16 | 77.69 | 55.50 | 61.20 | 69.21 | 84.51 | 1.973 | 0.083 |
| ViT-L/16+CE | 94.14 | 90.62 | 92.57 | 91.41 | 75.00 | 69.53 | 86.81 | 66.13 | 81.34 | | |
| RQ3: Performance comparison of ViTs+SLF and ViTs+NLL. | | | | | | | | | | | |
| ViT-L/16+SLF | 94.34 | 90.63 | 91.40 | 91.79 | 74.61 | 68.36 | 88.03 | 65.98 | 80.76 | 1.324 | 0.222 |
| ViT-L/16+NLL | 96.06 | 96.02 | 95.85 | 95.99 | 92.78 | 77.41 | 86.12 | 66.49 | 70.23 | | |
| RQ5: Performance comparison of ViT+SLF and ViT+SLF+H, the entropy regularization. | | | | | | | | | | | |
| ViT-L/16+SLF | 94.34 | 90.63 | 91.40 | 91.79 | 74.61 | 68.36 | 88.03 | 65.98 | 80.76 | 2.714 | 0.026 |
| ViT-L/16+SLF+$H$ | 95.31 | 95.7 | 95.12 | 94.53 | 91.99 | 74.8 | 89.16 | 69.18 | 81.89 | | |

Table 32: **RQ2:** Performance of CNN and ViT-L/16 backbones on CIFAR-10 and CIFAR-100 under closed-set label noise.

| Datasets | Arch. | Symmetric | | | Asymmetric | | | IDN-C | | BadLabel | | Avg. |
|---|---|---|---|---|---|---|---|---|---|---|---|---|
| | | 0.4 | 0.6 | 0.8 | 0.2 | 0.3 | 0.4 | 0.6 | 0.8 | 0.6 | 0.8 | |
| **CIFAR-10** | CNN | 58.19 | 38.75 | 19.09 | 83 | 78.15 | 73.69 | 52.22 | 28.04 | 35.66 | 13.44 | 48.02 |
| | ViT-L/16 | 81.35 | 57.23 | 25.6 | 93.38 | 88.36 | 79.19 | 44.29 | 22.12 | 38.28 | 16.79 | 54.66 |
| **CIFAR-100** | CNN | 40.72 | 22.98 | 7.55 | 58.25 | 50.3 | 41.53 | 52.55 | 40.45 | 17.05 | 4.18 | 33.56 |
| | ViT-L/16 | 67.05 | 51.94 | 27.86 | 77.21 | 67.83 | 56.24 | 57.42 | 35.54 | 30.46 | 11.33 | 48.29 |

Table 33: **RQ3:** Test Accuracy (mean±std) comparison for SLF and NLL methods on CIFAR-10 and CIFAR-100 datasets for ViT-B/16 backbone using MLP-K finetuning. Performance is reported for symmetric and asymmetric closed-set label noise for noise rate $\eta_{\text{sym}} \in \{0.4, 0.6, 0.8\}$ and $\eta_{\text{asym}} \in \{0.2, 0.3, 0.4\}$.

| | Method | Clean | Sym Noise Rate ($\eta$) | | | Asym Noise Rate ($\eta$) | | |
|---|---|---|---|---|---|---|---|---|
| | | | 0.4 | 0.6 | 0.8 | 0.2 | 0.3 | 0.4 |
| CIFAR-10 | CE | 96.09±0.02 | 81.35±0.23 | 57.23±0.43 | 25.60±0.41 | 93.38±0.11 | 88.36±0.21 | 79.19±0.39 |
| | FL | 95.70±0.01 | 82.68±0.27 | 58.48±0.59 | 26.32±0.55 | 89.84±0.09 | 87.89±0.30 | 81.24±0.24 |
| | GCE | 95.70±0.08 | 94.92±0.01 | 94.53±0.04 | 71.87±0.64 | 95.31±0.03 | 91.79±0.13 | 83.59±0.98 |
| | SCE | 95.31±0.06 | 94.53±0.19 | 84.37±0.25 | 39.77±0.35 | 94.14±0.01 | 89.45±0.17 | 83.59±0.48 |
| | NLNL | 95.74±0.13 | 80.67±0.09 | 23.08±0.12 | 10.04±0.52 | 92.51±0.10 | 84.74±0.12 | 80.63±0.13 |
| | DivideMix | 98.18±0.42 | 97.68±0.27 | 97.53±0.18 | 96.74±0.15 | 97.19±0.44 | 97.07±0.82 | 95.62±0.90 |
| | NCE+RCE | 95.70±0.06 | 95.31±0.08 | 93.75±0.04 | 88.67±0.14 | 95.70±0.04 | 95.31±0.07 | 93.35±0.27 |
| | NCE+AGCE | 94.53±0.05 | 93.75±0.06 | 93.70±0.10 | 90.62±0.46 | 96.09±0.09 | 96.09±0.07 | 94.99±0.41 |
| | ANL-CE | 95.70±0.55 | 95.31±0.37 | 95.05±0.18 | 93.75±0.58 | 95.70±0.31 | 94.53±0.32 | 91.66±0.76 |
| | Robust DivideMix | 98.09±0.19 | 97.56±0.41 | 97.43±0.63 | 96.55±0.36 | 97.23±0.29 | 97.19±0.98 | 95.68±0.75 |
| | CLIPCleaner | 96.35±0.12 | 95.32±0.18 | 95.23±0.24 | 94.89±0.55 | 95.10±0.43 | 95.24±0.47 | 95.26±0.73 |
| | NoiseGPT | 97.84±0.53 | 96.14±0.72 | 93.76±0.24 | 87.19±0.32 | 96.56±0.11 | 95.48±0.95 | 91.50±0.85 |
| CIFAR-100 | CE | 88.40±0.12 | 67.05±0.81 | 51.94±0.40 | 27.86±1.31 | 77.21±0.60 | 67.83±0.57 | 56.24±1.27 |
| | FL | 87.23±0.67 | 65.49±0.94 | 49.34±0.92 | 27.60±1.63 | 73.69±0.43 | 66.27±0.39 | 57.42±0.84 |
| | GCE | 88.15±0.48 | 87.62±0.48 | 85.54±0.84 | 78.64±1.28 | 87.23±0.74 | 77.59±0.29 | 60.02±0.63 |
| | SCE | 88.15±0.48 | 71.48±0.31 | 52.33±0.30 | 28.25±0.80 | 77.47±0.91 | 67.57±0.39 | 55.07±0.32 |
| | NLNL | 82.24±0.03 | 65.69±0.14 | 10.38±0.02 | 10.01±0.05 | 76.47±0.23 | 67.88±0.24 | 51.16±1.02 |
| | DivideMix | 87.12±0.32 | 80.54±0.09 | 72.59±0.18 | 55.54±0.23 | 86.66±0.27 | 85.64±0.42 | 80.12±0.85 |
| | NCE+RCE | 87.75±0.49 | 87.10±0.32 | 85.93±0.09 | 79.16±0.48 | 86.58±0.12 | 78.77±0.55 | 61.97±0.73 |
| | NCE+AGCE | 89.32±0.80 | 87.88±0.88 | 86.06±0.80 | 83.06±0.63 | 86.32±0.21 | 82.54±0.47 | 71.09±0.27 |
| | ANL-CE | 88.67±0.31 | 88.40±0.75 | 85.15±0.22 | 83.85±0.91 | 87.49±0.39 | 85.41±0.50 | 73.82±1.15 |
| | Robust DivideMix | 88.06±0.23 | 87.00±0.08 | 86.27±0.06 | 80.55±0.32 | 87.1±0.38 | 86.94±0.26 | 84.23±0.45 |
| | CLIPCleaner | 84.23±0.11 | 80.48±0.19 | 79.65±0.32 | 79.21±0.25 | 80.22±0.16 | 80.18±0.73 | 79.98±0.52 |
| | NoiseGPT | 82.79±0.10 | 78.87±0.12 | 71.75±0.54 | 64.25±0.73 | 80.56±0.85 | 79.87±0.39 | 79.18±0.22 |

Table 34: **RQ3:** Test Accuracy (mean±std) comparison for SLF and NLL methods on CIFAR-10 and CIFAR-100 datasets for ViT-B/16 backbone using MLP-K finetuning. Performance is reported for IDN-C and BadLabel noise for noise rate $\eta \in \{0.6, 0.8\}$.

| | Method | Clean | IDN-C Noise Rate ($\eta$) | | BadLabel Noise Rate ($\eta$) | |
| --- | --- | --- | --- | --- | --- | --- |
| | | | 0.6 | 0.8 | 0.6 | 0.8 |
| CIFAR-10 | CE | 96.09±0.02 | 44.29±0.52 | 22.12±0.17 | 38.28±0.24 | 16.79±0.32 |
| | FL | 95.70±0.01 | 44.92±0.44 | 18.35±0.18 | 39.06±0.63 | 16.79±0.72 |
| | GCE | 95.70±0.08 | 51.56±0.09 | 24.21±0.15 | 40.23±0.17 | 18.36±0.25 |
| | SCE | 95.31±0.06 | 46.87±0.13 | 20.7±0.32 | 39.45±0.45 | 16.41±0.11 |
| | NLNL | 95.74±0.13 | 68.35±0.16 | 31.64±0.44 | 40.23±0.73 | 16.4±0.95 |
| | DivideMix | 98.18±0.42 | 61.83±0.18 | 30.55±0.25 | 38.14±0.42 | 17.06±0.63 |
| | NCE+RCE | 95.70±0.06 | 52.34±0.17 | 21.09±0.19 | 44.53±0.22 | 17.18±0.82 |
| | NCE+AGCE | 94.53±0.05 | 54.29±0.45 | 19.92±0.19 | 46.09±0.24 | 16.79±0.32 |
| | ANL-CE | 95.70±0.55 | 53.91±0.52 | 20.31±0.10 | 40.62±0.15 | 14.45±0.34 |
| | Robust DivideMix | 98.09±0.19 | 60.89±0.72 | 28.26±0.85 | 41.85±0.09 | 17.72±0.22 |
| | CLIPCleaner | 96.35±0.12 | 94.48±0.64 | 77.21±0.44 | 95.3±0.63 | 84.87±0.57 |
| | NoiseGPT | 97.84±0.53 | 94.98±0.19 | 78.35±0.25 | 95.06±0.53 | 86.12±0.47 |
| CIFAR-100 | CE | 88.40±0.12 | 57.42±0.19 | 35.54±0.21 | 30.46±0.27 | 11.33±0.25 |
| | FL | 87.23±0.67 | 54.68±0.18 | 35.93±0.98 | 28.51±0.85 | 14.45±0.72 |
| | GCE | 88.15±0.48 | 59.37±0.47 | 33.59±0.35 | 26.56±0.58 | 10.54±0.68 |
| | SCE | 88.15±0.48 | 55.85±0.94 | 35.94±0.89 | 32.42±0.17 | 12.89±0.53 |
| | NLNL | 82.24±0.03 | 55.46±0.42 | 43.36±0.33 | 39.45±0.72 | 8.98±0.18 |
| | DivideMix | 87.12±0.32 | 62.42±0.22 | 39.42±0.16 | 39.89±0.15 | 18.01±0.14 |
| | NCE+RCE | 87.75±0.49 | 62.89±0.32 | 39.84±0.44 | 35.54±0.21 | 10.54±0.54 |
| | NCE+AGCE | 89.32±0.80 | 67.18±0.15 | 42.18±0.28 | 40.23±0.32 | 10.54±0.19 |
| | ANL-CE | 88.67±0.31 | 62.5±0.11 | 42.11±0.73 | 35.93±0.55 | 5.85±0.48 |
| | Robust DivideMix | 88.06±0.23 | 61.48±0.47 | 27.19±0.29 | 40.54±0.22 | 13.4±0.38 |
| | CLIPCleaner | 84.23±0.11 | 74.29±0.17 | 67.11±0.25 | 61.85±0.47 | 30.21±0.32 |
| | NoiseGPT | 82.79±0.10 | 76.64±0.63 | 67.28±0.95 | 61.96±0.85 | 31.84±1.09 |

Table 35: **RQ3:** Test Accuracy comparison for SLF and NLL methods on five real-world noisy labels datasets for ViT-L/16 backbone using MLP-K finetuning.

| Method | CIFAR10N | | | | | CIFAR100N | WebVision | Clothing1M | Food101N |
| --- | --- | --- | --- | --- | --- | --- | --- | --- | --- |
| | Aggregate | Random1 | Random2 | Random3 | Worst | | | | |
| CE | 94.14 | 90.62 | 92.57 | 91.41 | 75 | 69.53 | 86.81 | 66.13 | 81.34 |
| FL | 94.53 | 90.63 | 90.23 | 92.18 | 74.22 | 67.19 | 89.25 | 65.82 | 80.17 |
| GCE | 96.09 | 96.48 | 96.09 | 96.09 | 89.06 | 73.44 | 84.76 | 65.62 | 80.07 |
| SCE | 95.7 | 96.09 | 92.96 | 96.09 | 82.42 | 67.58 | 88.18 | 64.94 | 76.46 |
| NLNL | 96.48 | 96.09 | 95.7 | 96.48 | 94.92 | 75 | 86.03 | 67.67 | 12.35 |
| DivideMix | 97.12 | 97.12 | 97.12 | 97.12 | 97.08 | 78.45 | 80.28 | 65.68 | 65.51 |
| APL | 95.48 | 94.92 | 95.7 | 95.7 | 90.62 | 78.13 | 88.57 | 65.42 | 80.85 |
| NCE+AGCE | 96.09 | 94.92 | 96.87 | 95.7 | 92.96 | 80.08 | 88.37 | 64.84 | 81.15 |
| ANL | 96.48 | 96.48 | 96.09 | 94.92 | 94.14 | 79.29 | 89.06 | 64.94 | 80.17 |
| Robust DivideMix | 96.17 | 96.84 | 96.8 | 96.71 | 95.07 | 82.17 | 86.12 | 64.57 | 77.58 |
| CLIPCleaner | 95.26 | 95.3 | 95.21 | 95.22 | 95.11 | 78.13 | 84.72 | 69.35 | 74.24 |
| NoiseGPT | 95.77 | 95.98 | 95.92 | 95.87 | 96.39 | 81.81 | 85.12 | 71.84 | 73.95 |

Table 36: **RQ4:** Entropy Reduction and Robustness (Accuracy) across two standard classification losses and ten SOTA NLL methods using the ViT-L/16 backbone.

| Method | Closed-Set Noise | | Open-Set Noise | | Real-Wrold Noise | |
|---|---|---|---|---|---|---|
| | Acc. | Entropy Reduction | Acc. | Entropy Reduction | Acc. | Entropy Reduction |
| CE | 51.474 | 0.618 | 53.51 | 0.556 | 83.061 | 0.726 |
| FL | 50.948 | 0.553 | 55.727 | 0.623 | 82.691 | 0.789 |
| GCE | 63.654 | 0.825 | 74.607 | 0.872 | 86.411 | 0.911 |
| SCE | 54.928 | 0.427 | 56.117 | 0.652 | 84.491 | 0.704 |
| NLNL | 62.408 | 0.712 | 73.173 | 0.673 | 80.08 | 0.574 |
| DivideMix | 67.512 | 0.73 | 78.54 | 0.825 | 86.164 | 0.858 |
| APL | 66.278 | 0.85 | 74.997 | 0.844 | 87.266 | 0.856 |
| NCE+AGCE | 68.071 | 0.872 | 76.947 | 0.911 | 87.887 | 0.861 |
| ANL | 67.29 | 0.924 | 79.683 | 0.99 | 87.952 | 0.949 |
| Robust DivideMix | 69.253 | 0.708 | 79.49 | 0.696 | 88.003 | 0.794 |
| CLIPCleaner | 81.804 | 0.836 | 81.377 | 0.835 | 86.949 | 0.826 |
| NoiseGPT | 80.367 | 0.811 | 82.083 | 0.747 | 88.072 | 0.809 |

Table 37: **RQ4:** Prediction Entropy Reduction breakdown for CIFAR-10 and CIFAR-100 across two standard classification losses (SLF) and ten SOTA NLL methods using the ViT-L/16 backbone for closed-set noise settings.

| | Method | Symmetric Noise ($\eta$) | | | Asymmetric Noise ($\eta$) | | | IDN-C Noise ($\eta$) | | BadLabel Noise ($\eta$) | |
|---|---|---|---|---|---|---|---|---|---|---|---|
| | | 0.4 | 0.6 | 0.8 | 0.2 | 0.3 | 0.4 | 0.6 | 0.8 | 0.6 | 0.8 |
| CIFAR-10 | CE | 0.757 | 0.621 | 0.492 | 0.846 | 0.827 | 0.812 | 0.568 | 0.357 | 0.423 | 0.378 |
| | FL | 0.726 | 0.702 | 0.673 | 0.815 | 0.797 | 0.779 | 0.646 | 0.335 | 0.418 | 0.23 |
| | GCE | 0.943 | 0.889 | 0.355 | 0.931 | 0.857 | 0.783 | 0.835 | 0.804 | 0.812 | 0.757 |
| | SCE | 0.652 | 0.512 | 0.292 | 0.667 | 0.647 | 0.652 | 0.535 | 0.423 | 0.358 | 0.295 |
| | NLNL | 0.779 | 0.644 | 0.718 | 0.918 | 0.875 | 0.85 | 0.724 | 0.495 | 0.574 | 0.343 |
| | DivideMix | 0.957 | 0.949 | 0.936 | 0.946 | 0.914 | 0.882 | 0.591 | 0.479 | 0.533 | 0.362 |
| | APL | 0.967 | 0.961 | 0.958 | 0.969 | 0.954 | 0.918 | 0.857 | 0.809 | 0.679 | 0.428 |
| | NCE+AGCE | 0.964 | 0.962 | 0.959 | 0.966 | 0.96 | 0.932 | 0.885 | 0.832 | 0.746 | 0.443 |
| | ANL | 0.99 | 0.99 | 0.982 | 0.989 | 0.986 | 0.98 | 0.97 | 0.964 | 0.954 | 0.943 |
| | Robust DivideMix | 0.843 | 0.843 | 0.837 | 0.853 | 0.851 | 0.848 | 0.597 | 0.473 | 0.612 | 0.291 |
| | CLIPCleaner | 0.933 | 0.926 | 0.923 | 0.917 | 0.902 | 0.909 | 0.916 | 0.819 | 0.928 | 0.885 |
| | NoiseGPT | 0.797 | 0.767 | 0.786 | 0.843 | 0.821 | 0.794 | 0.825 | 0.673 | 0.814 | 0.54 |
| CIFAR-100 | CE | 0.724 | 0.66 | 0.383 | 0.855 | 0.836 | 0.729 | 0.75 | 0.426 | 0.513 | 0.41 |
| | FL | 0.721 | 0.571 | 0.362 | 0.737 | 0.619 | 0.509 | 0.527 | 0.402 | 0.301 | 0.191 |
| | GCE | 0.976 | 0.962 | 0.929 | 0.984 | 0.977 | 0.979 | 0.983 | 0.47 | 0.951 | 0.328 |
| | SCE | 0.435 | 0.271 | 0.124 | 0.576 | 0.487 | 0.435 | 0.496 | 0.429 | 0.158 | 0.092 |
| | NLNL | 0.865 | 0.832 | 0.791 | 0.891 | 0.849 | 0.757 | 0.789 | 0.692 | 0.582 | 0.279 |
| | DivideMix | 0.859 | 0.839 | 0.749 | 0.903 | 0.856 | 0.847 | 0.582 | 0.516 | 0.519 | 0.39 |
| | APL | 0.973 | 0.96 | 0.928 | 0.981 | 0.976 | 0.973 | 0.882 | 0.57 | 0.948 | 0.305 |
| | NCE+AGCE | 0.972 | 0.959 | 0.932 | 0.979 | 0.974 | 0.969 | 0.88 | 0.768 | 0.944 | 0.409 |
| | ANL | 0.995 | 0.993 | 0.988 | 0.996 | 0.995 | 0.993 | 0.896 | 0.694 | 0.992 | 0.188 |
| | Robust DivideMix | 0.854 | 0.842 | 0.839 | 0.886 | 0.85 | 0.8507 | 0.74 | 0.431 | 0.357 | 0.472 |
| | CLIPCleaner | 0.843 | 0.826 | 0.813 | 0.844 | 0.841 | 0.839 | 0.821 | 0.731 | 0.646 | 0.455 |
| | NoiseGPT | 0.964 | 0.935 | 0.921 | 0.945 | 0.923 | 0.916 | 0.872 | 0.831 | 0.782 | 0.463 |

## D.4 RQ4 RESULTS

Table 36 reports the test accuracy and corresponding prediction entropy reduction for twelve benchmarked methods across three noise categories: closed-set, open-set, and real-world label noise using ViT-L/16. To provide a more granular analysis, Table 37 details the entropy reduction results for closed-set noisy settings on the CIFAR-10 and CIFAR-100 datasets. Similarly, Table 17 shows the entropy reduction for the CIFAR80N-O dataset under open-set noise. Lastly, Table 38 presents the entropy reduction across all twelve methods on five real-world noisy datasets.

## D.5 RQ5 RESULTS

Table 29, 30, and 31 provides average performance for ViT-L/16+SLF and ViT-L/16+SLF+$H$. Tables 39, 39, and 40 provide a detailed breakdown of the performance improvements achieved by incorporating entropy regularization into the CE and FL loss functions across closed-set, open-set, and real-world noise categories for ViT-L/16. Across all settings, entropy regularization consistently enhances the robustness of ViTs to label noise.

Table 38: **RQ4:** Prediction Entropy Reduction breakdown for five real-world noisy label datasets across two standard classification losses (SLF) and ten SOTA NLL methods using the ViT-L/16 backbone.

| Method | CIFAR10N | | | | | CIFAR100N | WebVision | Clothing1M | Food101N |
|---|---|---|---|---|---|---|---|---|---|
| | Agg. | Rand1 | Rand2 | Rand3 | Worst | | | | |
| CE | 0.811 | 0.754 | 0.746 | 0.749 | 0.711 | 0.719 | 0.807 | 0.64 | 0.601 |
| FL | 0.878 | 0.872 | 0.869 | 0.855 | 0.691 | 0.717 | 0.903 | 0.684 | 0.636 |
| GCE | 0.98 | 0.973 | 0.973 | 0.972 | 0.899 | 0.979 | 0.948 | 0.604 | 0.867 |
| SCE | 0.938 | 0.894 | 0.89 | 0.889 | 0.747 | 0.984 | 0.134 | 0.335 | 0.522 |
| NLNL | 0.807 | 0.665 | 0.656 | 0.662 | 0.402 | 0.631 | 0.665 | 0.667 | 0.007 |
| DivideMix | 0.966 | 0.961 | 0.971 | 0.967 | 0.959 | 0.87 | 0.743 | 0.635 | 0.647 |
| APL | 0.993 | 0.989 | 0.99 | 0.99 | 0.96 | 0.981 | 0.956 | 0.075 | 0.767 |
| NCE+AGCE | 0.992 | 0.991 | 0.989 | 0.99 | 0.964 | 0.981 | 0.964 | 0.092 | 0.785 |
| ANL | 0.996 | 0.995 | 0.995 | 0.996 | 0.992 | 0.997 | 0.986 | 0.611 | 0.972 |
| Robust DivideMix | 0.855 | 0.868 | 0.866 | 0.866 | 0.864 | 0.774 | 0.757 | 0.628 | 0.668 |
| CLIPCleaner | 0.921 | 0.924 | 0.924 | 0.928 | 0.919 | 0.723 | 0.78 | 0.668 | 0.648 |
| NoiseGPT | 0.818 | 0.839 | 0.837 | 0.835 | 0.85 | 0.985 | 0.71 | 0.698 | 0.709 |

Table 39: **RQ5:** Impact of entropy regularization on performance for CIFAR-10 and CIFAR-100 under closed-set noisy settings using ViT-L/16 backbone.

| | Method | Symmetric Noise ($\eta$) | | | Asymmetric Noise ($\eta$) | | | IDN-C Noise ($\eta$) | | BadLabel Noise ($\eta$) | |
|---|---|---|---|---|---|---|---|---|---|---|---|
| | | 0.4 | 0.6 | 0.8 | 0.2 | 0.3 | 0.4 | 0.6 | 0.8 | 0.6 | 0.8 |
| CIFAR-10 | CE | 81.35 | 57.23 | 25.6 | 93.38 | 88.36 | 79.19 | 44.29 | 22.12 | 38.28 | 16.79 |
| | CE+$H$ | 96.48 | 96.48 | 95.7 | 96.48 | 88.67 | 82.03 | 66.01 | 26.56 | 43.75 | 23.05 |
| | | ↑15.13 | ↑39.25 | ↑70.10 | ↑3.10 | ↑0.31 | ↑2.84 | ↑21.72 | ↑4.44 | ↑5.47 | ↑6.26 |
| | FL | 94.92 | 94.14 | 93.14 | 65.41 | 65.31 | 59.16 | 46.48 | 13.28 | 24.22 | 6.25 |
| | FL+$H$ | 97.26 | 96.89 | 96.09 | 96.87 | 97.66 | 95.7 | 66.79 | 28.17 | 44.92 | 21.48 |
| | | ↑14.58 | ↑38.41 | ↑69.77 | ↑7.03 | ↑9.77 | ↑14.46 | ↑21.87 | ↑9.82 | ↑5.86 | ↑4.69 |
| CIFAR-100 | CE | 67.05 | 51.94 | 27.86 | 77.21 | 67.83 | 56.24 | 57.42 | 35.54 | 30.46 | 11.33 |
| | CE+$H$ | 85.54 | 83.2 | 78.51 | 83.2 | 76.17 | 68.75 | 64.45 | 47.26 | 37.89 | 16.41 |
| | | ↑18.49 | ↑31.26 | ↑50.65 | ↑5.99 | ↑8.34 | ↑12.51 | ↑7.03 | ↑11.72 | ↑7.43 | ↑5.08 |
| | FL | 65.49 | 49.34 | 27.6 | 73.69 | 66.27 | 57.42 | 54.68 | 35.93 | 28.51 | 14.45 |
| | FL+$H$ | 71.48 | 50.78 | 29.3 | 81.25 | 72.65 | 63.28 | 66.41 | 53.12 | 31.64 | 16.41 |
| | | ↑5.99 | ↑1.44 | ↑1.70 | ↑7.56 | ↑6.38 | ↑5.86 | ↑11.73 | ↑17.19 | ↑3.13 | ↑1.96 |

Table 40: **RQ5:** Impact of entropy regularization on performance for real-world noisy label datasets using ViT-L/16 backbone.

| Method | CIFAR10N | | | | | CIFAR100N | WebVision | Clothing1M | Food101N |
|---|---|---|---|---|---|---|---|---|---|
| | Agg. | Rand1 | Rand2 | Rand3 | Worst | | | | |
| CE | 94.14 | 90.62 | 92.57 | 91.41 | 75 | 69.53 | 86.81 | 66.13 | 81.34 |
| CE+$H$ | 94.92 | 95.7 | 94.92 | 94.53 | 91.8 | 73.82 | 88.18 | 68.43 | 82.23 |
| | ↑0.78 | ↑1.95 | ↑0.78 | ↑0.78 | ↑1.96 | ↑11.32 | ↑0.39 | ↑2.3 | ↑0.89 |
| FL | 94.53 | 90.63 | 90.23 | 92.18 | 74.22 | 67.19 | 89.25 | 65.82 | 80.17 |
| FL+$H$ | 95.7 | 95.7 | 95.31 | 94.53 | 92.19 | 75.78 | 90.13 | 69.92 | 81.54 |
| | ↑1.17 | ↑1.95 | ↑0.39 | ↑0.78 | ↑2.74 | ↑13.67 | ↑0.88 | ↑4.1 | ↑1.37 |

# E VIT-S/16 BACKBONE RESULTS

In this section, we present benchmarking results obtained with the ViT-S/16 backbone across two standard loss functions and ten noisy-label learning methods. Figure 9 summarizes the results under closed-set noise, including SLF+$H$. Results for open-set and real-world noise are provided in Figure 10.

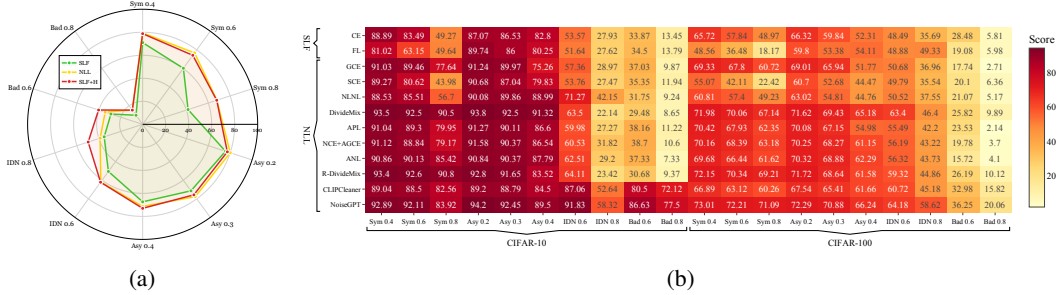

| | (a) | | (b) |

**Figure 9: RQ3:** a) Performance comparison of NLL-based methods under closed-set label noise on CIFAR-10 and CIFAR-100 using the ViT-S/16 backbone. b) Detailed results across datasets and noise settings.

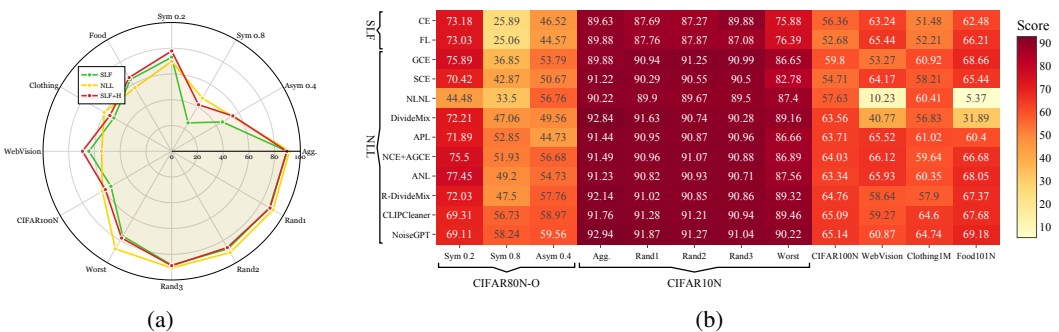

| | (a) | | (b) |

**Figure 10: RQ3:** a) Comparison of NLL-based methods using the ViT-S/16 backbone under open-set noise (CIFAR80N-O) and real-world noisy datasets: CIFAR-10N, CIFAR-100N, WebVision, Food101N, and Clothing1M. b) Detailed results across datasets and noise settings.

# F RESNET-50 BACKBONE RESULTS

We evaluate the ResNeT-50 backbone using two standard loss functions and ten state-of-the-art noisy-label learning methods. Figure 11 reports performance under closed-set noise, including results for SLF+$H$. Performance under open-set and real-world noise is summarized in Figure 12.

# G IMPORTANCE OF RECOMMENDED ENTROPY REGULARIZATION IN NOISE ROBUSTNESS

## G.1 GRADIENT-BASED INTUITION

**Cross-Entropy Gradient.** Cross-Entropy (CE) is widely used for training deep classifiers, however suffers from sensitivity to noisy labels. Given logits $z = (z_1, \ldots, z_k)$, the softmax probability for class $j$ is

$$p_j = \frac{e^{z_j}}{\sum_{\ell=1}^{k} e^{z_\ell}}.$$

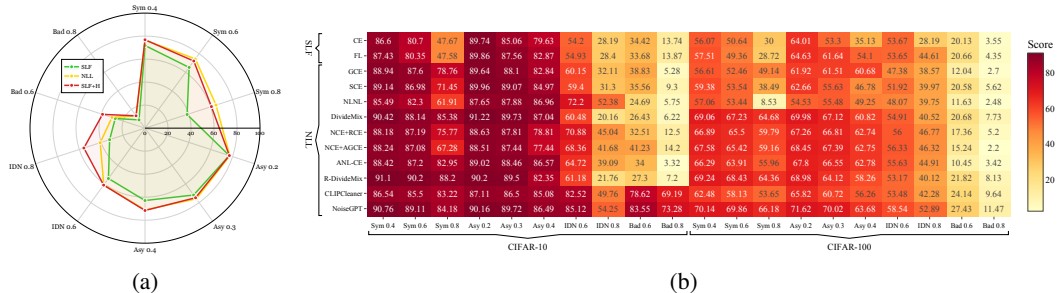

(a)                                                        (b)

Figure 11: **RQ3:** a) Performance comparison of NLL-based methods under closed-set label noise on CIFAR-10 and CIFAR-100 using the ResNet-50 backbone. b) Detailed results across datasets and noise settings.

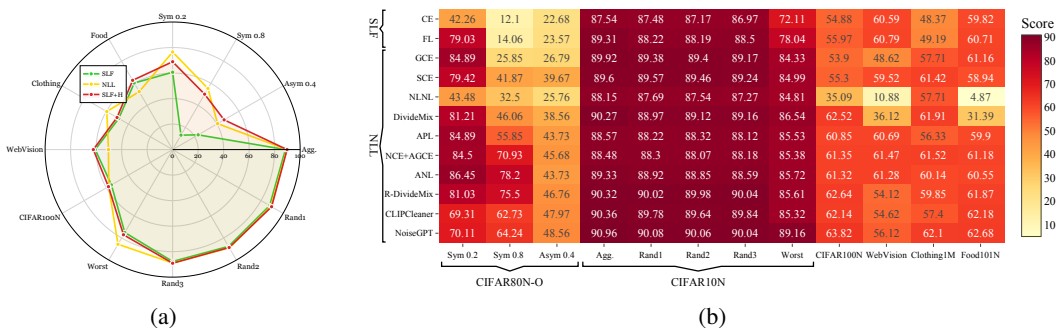

(a)                                                        (b)

Figure 12: **RQ3:** a) Comparison of NLL-based methods using the ResNet-50 backbone under open-set noise (CIFAR80N-O) and real-world noisy datasets: CIFAR-10N, CIFAR-100N, WebVision, Food101N, and Clothing1M. b) Detailed results across datasets and noise settings.

For a training sample with true label $y \in \{1, \ldots, k\}$, the cross-entropy loss is: $L_{\text{CE}}(z, y) = -\log p_y$. The gradient with respect to the logits $z_m$ is

$$\frac{\partial L_{\text{CE}}}{\partial z_m} = p_m - \delta_{ym},$$

where $\delta_{ym}$ is the *Kronecker delta*:

$$\delta_{ym} = \begin{cases} 1, & \text{if } y = m, \\ 0, & \text{if } y \neq m. \end{cases}$$

Equivalently, in vector form, $\nabla_{z_j} L_{\text{CE}}(p, \tilde{y}) = p_j - \mathbf{1}\{j = \tilde{y}\}$. For a mislabeled example where $p_{\tilde{y}}$ is small, the gradient magnitude of CE is large and points towards fitting the wrong label. This makes CE prone to overfitting noisy samples.

**Entropy Regularizer Gradient.** The (Shannon) entropy of the predictive distribution $p$ is

$$H(p) = -\sum_{j=1}^{k} p_j \log p_j.$$

For an entropy-regularized loss with weight $\lambda$, the gradient with respect to $z_m$ is

$$\nabla_{z_m}(\lambda H) = \lambda p_m \left( \sum_{j=1}^{k} p_j(\log p_j + 1) - (\log p_m + 1) \right).$$

**Combined Loss.** Thus, for a cross-entropy loss with entropy regularization,

$$L_{\text{CE+H}} = L_{\text{CE}} + \lambda H,$$

the gradient with respect to the logits is

$$\nabla_{z_m} L_{\text{CE+H}} = \big(p_m - \delta_{ym}\big) + \lambda p_m \left( \sum_{j=1}^{k} p_j (\log p_j + 1) - (\log p_m + 1) \right).$$

## G.2 WORKED EXAMPLE ON CIFAR-10

We illustrate with a toy example on CIFAR-10 ($k = 10$ classes). Suppose the clean label is "cat" but due to symmetric noise it is flipped to "dog". The network predicts the following probabilities:

$$p = [0.05, 0.10, 0.05, 0.55, 0.05, 0.05, 0.05, 0.05, 0.00, 0.00],$$

where $p_{\text{cat}} = 0.55$ and $p_{\text{dog}} = 0.05$.

**Case 1: CE.** With noisy label $\tilde{y} = \text{dog}$, the loss is:

$$L_{\text{CE}} = -\log p_{\text{dog}} = -\log(0.05) \approx 3.00.$$

and Gradient is:

$$\nabla_{z_{\text{dog}}} L_{\text{CE}} = 0.05 - 1 = -0.95, \qquad \nabla_{z_{\text{cat}}} L_{\text{CE}} = 0.55 - 0 = 0.55.$$

Although the model predicts the true class correctly, CE strongly pushes the network toward the *noisy* label.

**Case 2: CE+H.** Entropy term ($\lambda = 0.2$):

$$\nabla_{z_{\text{cat}}} L_{\text{CE+H}} = 0.55 + \lambda 0.55 \left( 0.302 - (-1.30 + 1) \right) = 1.071$$

$$\nabla_{z_{\text{Dog}}} L_{\text{CE+H}} = -0.95 + \lambda 0.05 \left( 0.302 - (-0.259 + 1) \right) = -0.637$$

The entropy regularization term has reduced magnitude of derivative of noisy label i.e. Dog from 0.95 to 0.637. while the magnitude of the derivative of clean but unknown label cat has increased from 0.55 to 1.071. Therefore, when the noisy label pushes the model away from the clean label the entropy regularizer act as a damping factor and pushes the model towards clean but unknown label. As a result, mislabeled samples no longer produce extreme gradients, making the training dynamics more robust under noise. Thus, CE+H has better robustness than CE alone and CE+H balances the gradient updates.

