# OpenReview forum: "ViT Fine-tuning Vulnerability to Label Noise"
_ICLR.cc/2026/Conference — ICLR 2026 Conference Withdrawn Submission_

### Official Review · Reviewer_BwTi · 2025-10-18

**Soundness:** 3
**Presentation:** 3
**Contribution:** 3
**Rating:** 6
**Confidence:** 4

**Summary:**

This paper presents a comprehensive empirical study of how label noise affects Vision Transformers (ViTs) and examines the effectiveness of learning-with-noisy-labels (NLL) techniques. The authors formulate five research questions with corresponding hypotheses and design targeted experiments to test each one. The principal finding is that, during early training, ViTs are less vulnerable to label noise than CNN-based models. Moreover, established NLL methods mitigate performance degradation when the corruption is closed-set. The paper also proposes adding an entropy regularization term to the training objective, further improving ViT robustness to label noise.

**Strengths:**

- The manuscript is clearly written, well structured, and easy to follow; the motivation is explicit, and the experiments/visualizations convincingly support the claims.
- The research design is sound, and organizing the study around explicit research questions greatly improves readability.
- The experimental evaluation is extensive, covering diverse scenarios with relevant datasets and architectures.
- The observed correlation between entropy reduction and model performance is insightful and could spur further investigation.
- The systematic exploration of fine-tuning strategies under noisy labels offers practical value to the community by clarifying how different techniques behave in this setting.

**Weaknesses:**

- From-scratch training and CNN–ViT parity under NLL are missing. Most NLL techniques were developed and validated for CNNs in full-training regimes. Because the study centers on fine-tuning, it remains unclear whether ViTs benefit from NLL to the same extent as CNNs when trained from scratch. A controlled comparison—CNN+NLL vs. ViT+NLL, trained end-to-end on the same datasets and noise rates—would offer a more complete picture of ViT vulnerability to label noise and clarify how architectural differences interact with NLL methods.

- Entropy regularization and entropy dynamics need deeper justification and broader validation. The proposed entropy term would benefit from clearer intuition for why it improves robustness under noisy labels, evidence of whether the effect transfers to CNN-based architectures, and discussion of applicability beyond vision tasks. Relatedly, the entropy-reduction analysis feels underdeveloped: please report early-epoch (e.g., after the first epoch) predictive entropy across models and noise levels, examine whether performance gains correlate with initial entropy or with its decay over training, and articulate a plausible mechanism linking entropy dynamics to the observed improvements.

**Questions:**

- The study centers on fine-tuning, while most NLL methods were designed for full training on CNNs. How would you expect the results to change in a _from-scratch_ regime? Specifically, in a head-to-head setting (same datasets/noise rates/training budgets), would ViT+NLL match or surpass CNN+NLL?
- Could you provide intuition or a theoretical sketch explaining why the entropy term improves robustness under noisy labels? How sensitive are gains to the regularizer weight, and how does it compare to related techniques (e.g., label smoothing, confidence penalty, temperature scaling)? Does the effect transfer to CNNs, and to non-vision domains?
- The study notes that NLL methods help primarily for closed-set noise. How could the proposed approach and baselines improve their performance under open-set noise in terms? How do they behave in that scenario in terms of predictions and noise detection? Are there regimes where ViTs lose their early robustness advantage or where the entropy term harms learning?

---

### Official Review · Reviewer_S5DU · 2025-10-25

**Soundness:** 1
**Presentation:** 2
**Contribution:** 2
**Rating:** 2
**Confidence:** 4

**Summary:**

This paper introduces NLL-ViT, a comprehensive benchmark studying the robustness of Vision Transformer (ViT) fine-tuning under label noise. It evaluates two ViT backbones (ViT-B/16, ViT-L/16), multiple fine-tuning strategies (Full-FT, AdaptFormer, VPT, MLP-K, Linear Probing), two standard classification losses (Cross-Entropy, Focal), and ten state-of-the-art noisy label learning (NLL) methods originally proposed for CNNs across three noise categories (closed-set, open-set, real-world) and eight datasets (CIFAR-10/100, CIFAR-10N/100N, CIFAR80N-O, WebVision, Clothing1M, Food101N).

**Strengths:**

1. The research content of this paper is highly valuable and closely aligns with practical training and application methods. It is worthy of in-depth study.
2. Covers three noise categories (closed-/open-set/real-world), eight datasets, two ViT scales (plus supporting ViT-S/16), five fine-tuning paradigms, and ten NLL methods.
3. The author conducted a large number of comprehensive experiments, which is worthy of praise.

**Weaknesses:**

1. The real-world dataset may also contain a portion of the open-set noise, such as WebVision. This is also worthy of distinction and discussion.
2. The author claims that the existing methods are ineffective on the open dataset. However, Table 13 shows that many of these methods are still effective.
3. The author claimed to have conducted three repeated experiments. However, some of the experimental results had standard deviations while others did not, which made it quite confusing. Were all the experiments repeated three times?
4. Minor problem: Some confusing citations, such as IDN-C Ye et al. (2023) in 182 line.
5. The author's claim that ViT is more robust than CNN is not rigorous. Because the suitability of different architectures for training parameters may vary, for example, learning rate, weight decay, optimizer, etc. The good results of ViT might merely be due to the fact that the training parameters used are more suitable for ViT rather than for CNN. To prove that ViT is more robust than CNN requires more detailed experimental comparisons.
6. A noisy label learning algorithm [1] similar to the entropy regularization in this paper has already been proposed.
7. The hyperparameter choices for different methods appear inappropriate. Most noisy label learning methods are sensitive to hyperparameters. If the experimental setup does not fully match that of the original papers, directly reusing their hyperparameters is not suitable. Under different setups—datasets, architectures, or training settings—the hyperparameters for each method can vary. A more rigorous approach is to select suitable hyperparameters for all baselines under the current experimental conditions. For example, in Figure 4, DivideMix shows extremely poor performance, which is implausible given prior papers. Similarly, in Table 11, GCE is reported at 6% (which may be a typo), while CE reaches 86%, an implausible gap under typical conditions. These discrepancies further suggest that the baselines were not properly tuned.

Overall, I think the research direction of this paper is quite meaningful, but the specific implementation requires significant improvements.

[1] Learning with Noisy Labels via Sparse Regularization. ICCV 2021.

**Questions:**

Why wasn't this paper submitted to the "datasets and benchmarks" topic?

---

### Official Review · Reviewer_iNCz · 2025-10-31

**Soundness:** 1
**Presentation:** 1
**Contribution:** 1
**Rating:** 2
**Confidence:** 5

**Summary:**

This paper presents a limited scientific contribution. It primarily benchmarks the performance of Vision Transformers (ViTs) under various label-noise settings, comparing them with CNNs and applying existing noisy-label learning (NLL) techniques. While the experimental scale is extensive, the paper introduces no novel methodology, theoretical insight, or meaningful algorithmic advancement. The writing is often unclear and difficult to follow, and the overall impact is limited to reporting empirical observations.

The entropic analysis at the end offers a minor empirical curiosity but is insufficient for publication at this level. Moreover, several choices are unjustified—for example, the use of Focal Loss [Lin 2017], which is designed for class-imbalance scenarios, is not explained. The paper also suffers from poor writing quality: cross-entropy is introduced three times with its acronym defined inconsistently, and several other acronyms are redundantly reintroduced.

In lines 91–93, a list of methods is shown without any references, while the same list -with citations - is repeated between lines 214–215 and elsewhere. Much of the introduction is repeated almost verbatim in the Related Work section. Tables are difficult to interpret and stylistically inconsistent. “Finetuning” is misspelled multiple times throughout the paper, and some plots (e.g., Figure 3) are hard to read. The citations are not wrapped between parentheses when it's required.

**Strengths:**

The paper provides an extensive empirical benchmark on the robustness of Vision Transformers
(ViTs) to label noise.

**Weaknesses:**

This paper presents a limited scientific contribution. Its main focus is benchmarking the performance of Vision Transformers (ViTs) under various label-noise settings, comparing them with CNNs and applying existing noisy-label learning (NLL) techniques. While the experimental scale is extensive, the paper introduces no novel methodology, theoretical insight, or meaningful algorithmic advancement. The proposed entropy regularization is simplistic, purely empirical, and lacks theoretical justification.

Several methodological choices are unclear or unjustified; for example, the use of Focal Loss [Lin 2017], which is designed for class-imbalance scenarios, is never explained or motivated. The observed robustness of ViTs to label noise is simply reported, without any ablations or architectural reasoning to support it.

The writing is often unclear and difficult to follow, with numerous repetitions and inconsistencies. Specifically, cross-entropy is introduced three times with its acronym defined differently each time, and several other abbreviations are redundantly reintroduced. In lines 91–93, a list of methods is shown without any references, while the same list—with citations—is repeated between lines 214–215 and again elsewhere. Much of the introduction is repeated almost verbatim in the Related Work section. Tables are difficult to interpret and stylistically inconsistent across the paper; for example, numerical formatting and column alignment vary. Figures (especially Figures 3 and 4) are dense and hard to read. The term “Finetuning” is misspelled multiple times, and citations are not wrapped in parentheses where required.

Overall, while appreciating the experimental effort, the work reads more like an internal benchmark report than a scientific contribution. It fails to explain why ViTs appear more robust to label noise than CNNs, offering no architectural analysis, theoretical insight, or attention-based interpretation.

**Questions:**

Why was Focal Loss chosen for benchmarking despite its design for class imbalance rather than
label noise?

---

### Official Review · Reviewer_7YvQ · 2025-11-04

**Soundness:** 2
**Presentation:** 2
**Contribution:** 2
**Rating:** 4
**Confidence:** 4

**Summary:**

This paper presents a comprehensive benchmark study of Vision Transformers' (ViTs) robustness to noisy labels. Through 850+ experiments across 8 datasets and 3 noise types (closed-set, open-set, real-world), the authors find that ViTs are more robust to label noise than CNNs, but existing noisy label learning methods designed for CNNs can fail on open-set and real-world noise. The paper proposes entropy-based regularization that significantly improves ViT robustness across all noise settings.

**Strengths:**

- The paper studies an important problem of learning with noisy labels.
- Extensive experiments are done with a wide range of datasets, and goes beyond synthetic closed-set noise to realistic scenarios.
- The authors of the paper suggest a simple entropy regularization that practitioners can adopt.

**Weaknesses:**

- I am not sure if the claim that entropy regularization is preferred over other existing NLL methods is valid. To the best of my understanding, the authors take average across all previously proposed NLL methods. This doesn't seem like a fair comparison. Why don't the authors compare against each method individually?

- While comprehensive experiments were conducted, the authors of the paper only carry out using  ViT-S/16, -B/16 -L/16. Since the paper is empirically driven, it would be interesting to also look at other more recent vision transformer models.

- For RQ2, what is the CNN used? What is the performance of the CNN model on clean dataset? Is the CNN model pre-trained? If not, how do you know the "robustness" comes from the architecture or pretraining? prior works [1] have demonstrated that pretraining makes model more robust to noise. In general, it is unclear what explains why CNN-based NLL methods fail on ViTs in open-set/real-world settings. Deeper analysis needed.

All in all, despite the extensive experiments, I find some of the conclusions drawn somewhat unconvincing.

[1] "Using Pre-Training Can Improve Model Robustness and Uncertainty"

**Questions:**

- For Noisy label learning methods, why do you take the average across different methods? Is there a particular method that performs well across different datasets? How does the best performing method compare with entropy regularization?

---

### Note · Authors · 2025-11-23

I have read and agree with the venue's withdrawal policy on behalf of myself and my co-authors.